# Assaying Out-Of-Distribution Generalization in Transfer Learning

**Florian Wenzel**[* 1]  **Andrea Dittadi**[† 2]  **Peter Gehler**[1]
**Carl-Johann Simon-Gabriel**[1]  **Max Horn**[1]  **Dominik Zietlow**[1]
**David Kernert**[1]  **Chris Russell**[1]  **Thomas Brox**[1]
**Bernt Schiele**[1]  **Bernhard Schölkopf**[1]  **Francesco Locatello**[1]

[1] AWS Tübingen [2] Technical University of Denmark

## Abstract

Since out-of-distribution generalization is a generally ill-posed problem, various proxy targets (e.g., calibration, adversarial robustness, algorithmic corruptions, invariance across shifts) were studied across different research programs resulting in different recommendations. While sharing the same aspirational goal, these approaches have never been tested under the same experimental conditions on real data. In this paper, we take a unified view of previous work, highlighting message discrepancies that we address empirically, and providing recommendations on how to measure the robustness of a model and how to improve it. To this end, we collect 172 publicly available dataset pairs for training and out-of-distribution evaluation of accuracy, calibration error, adversarial attacks, environment invariance, and synthetic corruptions. We fine-tune over 31k networks, from nine different architectures in the many- and few-shot setting. Our findings confirm that in- and out-of-distribution accuracies tend to increase jointly, but show that their relation is largely dataset-dependent, and in general more nuanced and more complex than posited by previous, smaller scale studies[1].

## 1 Introduction

With deep learning enabling a variety of downstream applications [1–4], failures of robustness leading to systematic [5–7] and catastrophic deployment errors [8–10] have become increasingly relevant. From early work on studying distribution shifts [e.g., 11, 12] and the classical "cow on the beach" example (e.g., in [13]), several works have highlighted sometimes spectacular failures of machine learning when the test distribution differs from training [10, 14–21]. This has motivated the study of different types of distribution shifts, ultimately branching the field into several sub-communities that, while sharing the same underlying objective, rely on different evaluation protocols and provide different recommendations to practitioners.

**(1)** The studies [15, 17–20, 22, 23] focused on algorithmically corrupting upstream pre-training datasets [24] to test generalization. Perhaps unsurprisingly, the choice of augmentations can significantly alter this notion of robustness [25–29]. **(2)** As synthetic corruptions need not transfer to real world distribution shifts [27], new realistic datasets were collected to test upstream robustness [15, 27, 30–33]. Here, scale has been identified as a reliable ingredient [30, 34–37], despite

---

[*]Correspondence to: `flwenzel@amazon.de`.

[†]Part of this work was done during an internship at AWS Tübingen.

[1]The code for the evaluation study is at github.com/amazon-research/assaying-ood. Author contributions are listed at the end of paper.

36th Conference on Neural Information Processing Systems (NeurIPS 2022).

other works [38] arguing that extensive upstream pre-training can harm downstream robustness. **(3)** Exhaustive comparisons attempted to disentangle intrinsic architectural robustness from specific training schedules [39–44], addressing underspecification [45] with inductive biases. Orthogonally, several (less scalable) works advocated for leveraging the compositional (perhaps causal [46]) structure in the underlying data-generative process to introduce suitable inductive biases [47–53]. **(4)** Simultaneously, Bayesian approaches for uncertainty predictions have been proposed to improve model calibration [54–61] and robustness on new distributions [62–64]. Recent work, however, found that larger models were natively better calibrated [65]. **(5)** The adversarial training community developed an entire literature on different worst case local perturbations of training data [14, 66], with 5000+ papers written to date [67] and a never ending cycle of new defenses and attacks [68–72]. **(6)** Other niche approaches investigated carefully designed test sets [73–75] and training protocols that promote invariance across several distributions [73, 76–79]. Despite this progress, *empirical risk minimization (ERM)* remains a strong contender [75]. Overall, the significant community effort towards more robust machine learning models have resulted in diverse proxy evaluation targets yielding different practical recommendations.

At the same time, the workflow of successful applications developed in the opposite direction [3, 4, 80–82]. Instead of collecting large application-specific datasets, one trains generalist backbones on the greatest possible amount of data and then transfers the model using available domain-specific examples. Besides the test data likely being "on manifold", one is almost certainly guaranteed that there will be some sort of distribution shift at test time as the size of the fine-tuning dataset decreases.

Focusing on classification of visual data, we evaluate the different key metrics from these communities in a unified manner and under the same experimental conditions to investigate the gaps in common practices. We restrict ourselves to the realistic situation where we have an ImageNet pre-trained model available and a new target distribution as downstream task. After the model has been fine-tuned, the test data may be OOD. From 36 existing datasets, we extract 172 *in-distribution (ID)* and *out-of-distribution (OOD)* dataset pairs, fine-tuning and evaluating over 31k models to gain a broader insight in the sometimes contradicting statements on OOD robustness in previous research. We organize our study around two key questions: (1) What are good proxy measures of OOD robustness when having access to a single dataset? (2) How do architecture choices and fine-tuning strategies affect robustness? We plan to publish the code with the camera-ready version of the paper.

Our key contributions are **(1)** We conduct a large systematic study of OOD robustness, evaluating the effect of architecture type, augmentation, fine-tuning strategies and few-shot learning. We investigate the interplay of robustness to corruptions, adversarial robustness, robustness to natural distribution shifts, calibration and other robustness metrics in a unified setting and under the same experimental conditions. **(2)** We find that out-of-distribution generalization has many facets. Insights of previous papers—sometimes presented as general conclusions— hold only on a subset of the tasks/datasets included in our study and hence actually only reflect a special case. **(3)** In general, in-distribution classification error (accuracy) is the best predictor of OOD accuracy, but other secondary metrics can provide additional insights. **(4)** With these results, we revisit previous studies and recommendations, reinterpreting their conclusions, resolving some contradictions, and suggesting critical areas for further research.

## 2 Experimental setup

We follow the modern workflow of applications of computer vision to (long-tail) downstream tasks from existing pre-trained backbones. The model is transferred using a set of (potentially few) examples from a new distribution. At test time, we assume that the classes remain the same (closed-world setting), but that the distribution may otherwise change. We specifically focus on the effect of distribution shifts *after* a model has been transferred to a new distribution (i.e., the *downstream* implications) and discuss the empirical differences and similarities compared to results concerning *upstream* OOD robustness that were discussed in previous studies [15, 27, 30–34].

*Experimental protocol and datasets:* We evaluate nine state-of-the-art deep learning models with publicly available pre-trained weights for ImageNet1k / ILSVRC2012 [24]. We consider 36 datasets grouped into ten different *tasks* sharing the same labels. Datasets of the same task represent a set of natural distribution shifts. For each task, we take a single training dataset to fine-tune the model and report evaluation metrics on both its ID test set and all the other OOD test sets. We

extract 172 (ID, OOD) dataset pairs from the different domains of the ten tasks: DomainNet [83], PACS [84], SVIRO [85], Terra Incognita [13] as well as the Caltech101 [86], VLCS [87], Sun09 [88], VOC2007 [89] and the Wilds datasets [90] (from which we extract two tasks). In our experimental protocol we do not make any assumptions on the particular shift type and the considered tasks reflect multiple shift types (e.g., presumably a strong covariate shift in DomainNet and a partial label shift in Camelyon17 of the Wilds benchmark). See Appendix E for a detailed overview. Models are fine-tuned on a single GPU using Adam [91] with a batch-size of 64 and a constant learning rate.

*Evaluation on ID, OOD and corrupted data:* Some tasks, such as DomainNet, PACS and SVIRO come with different datasets/domains. For those, we report for each dataset the ID (test) performance and the OOD (test) performances on the other datasets in the task. For the datasets from the WILDS benchmark, we use the provided ID test and OOD test splits. If a task consists of multiple OOD data we compute the metrics additionally on held-out OOD data. To do so, for each (ID, OOD) dataset pair, we average the performance on the remaining OOD datasets. This approach is sometimes called multi-domain evaluation [e.g., 64]. Alongside the provided OOD datasets, we evaluate the models on the corrupted ID test set. We apply 17 types of corruptions from [17] each with 5 severity levels. The corrupted version of the datasets can be viewed as a synthetic distribution shift and we investigate how informative they are of natural distribution shifts.

*Models:* To ensure that our results are relevant for researchers and practitioners alike, we consider both widely deployed and recent top-performing methods: Resnet50d [92], DenseNet [93], EfficientNetV2 [94], gMLP [95], MLP-Mixer [96], ResMLP [97], Vision Transformers[1] [4], Deit [98], Swin Transformer [99]. We list the exact model names in Table S4. Our choice of models covers convolutional networks, transformer variants and mixers. Weights for the pre-trained models were taken from the PyTorch Image Models repository [100].

*Model hyperparameters and augmentation strategies:* For each model we consider the learning rate and the number of fine-tuning epochs. We first ran a large sweep over these two hyperparameters on a subset of the experiments and used it to pre-select a set of four parameter combinations that included the best performing models for each architecture. Additionally, we study three different augmentation strategies: standard ImageNet augmentation (i.e., no additional augmentation), *RandAugment* [101] and *AugMix* [28]. More details can be found in Appendix H.

*Fine-tuning strategy and few-shot training:* We investigate fine-tuning the full architecture and fine-tuning only the head. Additionally, we consider three training paradigms: training on the full downstream dataset, and two few-shot settings: "few-shot-100" (a subset with 100 examples per class, if available) and "few-shot-10" (with 10 examples per class). In the few-shot settings, the images are randomly selected, and classes that have fewer images as the cap of 10 or 100, respectively, are not over-sampled.

*Metrics:* We pick some of the most popular metrics that are used to measure progress towards robust machine learning. We report six different metrics: *classification error*, *negative log-likelihood (NLL)*, *demographic disparity* [102, 103] on inferred groups [77] as a measure of invariance[2], the *expected calibration error (ECE)* [104], and adversarial classification error for two different $\ell_2$-attack sizes. The metrics are, where applicable, evaluated on ID, OOD, and corrupted test sets, (except adversarial error, which we did not evaluate on the corrupted test sets). See Appendix G for more details.

## 3  Additional related work

As much of the related work was already mentioned in the introduction we highlight two main areas of closely related works: one regarding benchmarks for generalization to new distributions and one on the interplay between different evaluation metrics.

**Benchmarking robustness to OOD.** Closely to our setting, [105] benchmarked models in a few-shot learning setting but did not analyze the robustness of the fine-tuned models. In follow up work, [34] related the results of [105] to *upstream* robustness but did not consider downstream distribution shifts. [75] analyzed a variety of domain generalization algorithm and found that none of them could beat a strong ERM baseline. While several of our datasets overlap with theirs, we consider the transfer

---

[1]Trained on ImageNet21k and fine tuned on ImageNet1k.

[2]As there is no "measure of invariance" for a single dataset, we rely on [77] that finds a partition of the data maximising the IRM [76] penalty.

learning setting as opposed to domain generalization. Their insights may be in part explained by the fact that the regularization is either orthogonal to OOD accuracy or simply harms accuracy overall as in [106]. [21, 107] proposed a model for analyzing different fine-grained distribution shifts. Their work is limited to few datasets and model types and only cover accuracy evaluations. [108] studied the effect of the pre-training strategy on domain generalization and [109] studied extensions to large pre-trained models for improved reliability, whereas our work analyzes fine-tuning protocols and the robustness on downstream tasks. [27] found that larger models and better augmentation techniques improve robustness but did not consider different model types, augmentation techniques or evaluation metrics. Our work studies robustness in a larger scope than previous work, which focused on certain dimensions of our empirical investigations. None of the previous work studied the interplay of different robustness metrics.

**Studying the interplay of robustness metrics.** There has been only limited work on analyzing informativeness of robustness metrics on OOD generalization. [110] analyzed distribution shifts of ImageNet and found that corruption metrics do not imply robustness to natural shift. Recently, [35], based on previous studies by [15, 110–112], observed a clear linear relationship between ID accuracy and OOD accuracy and hypothesized that this could be a general pattern in contradiction to [45]. [113] extended this line of work to agreement between networks and [114] found that large pre-trained models are above the linear trend in early stages of fine-tuning. However, our extended set of experiments show that a clear linear trend is only visible on some (ID, OOD) dataset pairings. [115] empirically investigated different generalization measures and found that measures relating to the Fisher information perform best.

# 4 A broad look at out-of-distribution generalization

In the following we explore the facets of out-of-distribution generalization, highlighting discrepancies to prior work and discuss their implications.

## 4.1 The main latent factors that explain the empirical results

To get a first overview of the relations between the different metrics and their generalization properties, we perform a factor analysis to discover the main orthogonal latent factors that explain the variance in the metrics evaluated on each ID dataset, its corrupted variant, and the metrics averaged over all compatible OOD datasets for each fine-tuned model. For details, see Appendix B.

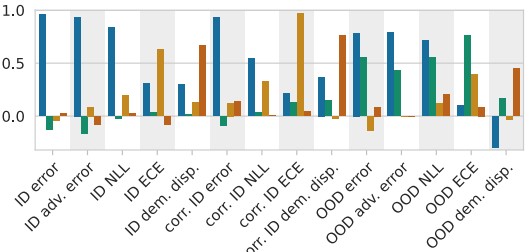

Based on the scree plot in Appendix B, we retain four factors. Their contributions (loadings) to each metric are shown in Fig. 1. Interestingly, each factor has a clear interpretation. Factor 1 (blue) is very well aligned with ID classification error, log-likelihood and adversarial attacks. Factor 2 (green) captures OOD-specific

Figure 1: Factor loadings (contributions) of different metrics based on a factor analysis with 4 orthogonal factors (color-coded), highlighting similarities between the metrics. The factor **Blue**: captures classification error, adversarial error, log-likelihood, and their corrupted variants. **Green**: only in OOD metrics. **Yellow**: expected calibration error. **Red**: demographic disparity.

variance, since it is particularly pronounced in almost every out-of-distribution metric, and only there. Factor 3 (orange) relates mainly to the expected calibration error and factor 4 (red) to demographic disparity. The dominant presence of factor 1 (blue) in all classification error and log-likelihood metrics ID and OOD suggests that ID classification error can be a reasonably good predictor of OOD classification error, which we further discuss in Section 4.3. However, the presence of an OOD-specific factor also suggests that ID versus OOD accuracy (classification error) cannot always lie "on a line" [35]—we investigate this further in Section 4.2. Another noteworthy point is that the corrupted metrics and adversarial classification errors have almost no OOD component and are generally very close to the corresponding ID metric. Similarly, the loadings of the corrupted metrics are much closer to those of the ID metrics than to OOD metrics. This suggests that the performance on artificially corrupted data may not predict the OOD performance significantly better than the bare ID metrics. We further discuss this in Section 4.3.2. Finally, the fact that demographic disparity and

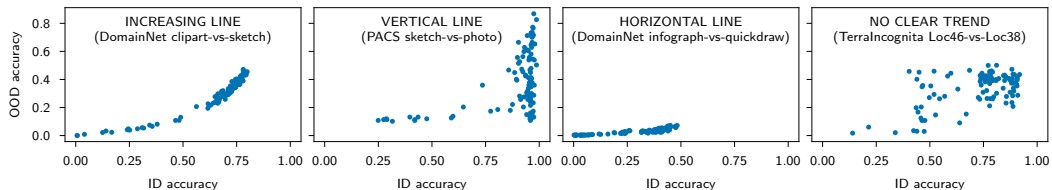

Figure 2: Typical scatter plot patterns observed in our data (see Appendix D for all plots). *Increasing line*: ID and OOD accuracy show a clear functional dependency. In contrast to previous claims this is not the typical setting (only observed on a subset of datasets). *Vertical line*: the same ID peformance leads to different OOD behavior (underspecification setting). *Horizontal line close to zero accuracy*: no transfer of information from the ID to the OOD dataset. *No clear trend*: random associations between ID and OOD accuracy (i.e., zero correlation).

expected calibration error are each mainly captured by their own, specific factor suggests that, maybe surprisingly, those metrics are largely independent of the networks' classification error. Further details are discussed in Section 4.3.2.

> **Takeaway**: One latent factor suffices to capture accuracy and log-likelihood on ID, corrupted, and adversarial datasets. OOD behavior, calibration, and environment invariance are each captured by a separate factor. A separate factor for OOD metrics suggests that artificial and adversarial corruptions do not fully mimic real distribution shifts.

## 4.2 The many facets of out-of-distribution generalization

Prior publications [34, 35, 110] observed that OOD accuracy strongly linearly correlates with ID accuracy, or, in other words, that ID vs OOD accuracy nearly lie "on a line". In contrast, we find that this is not a general trend when tested on more tasks. Fig. 2 shows the four typical settings we observe. For some (ID, OOD) dataset pairs we observe a clear functional dependency as claimed by [35, 110] (increasing line). For other dataset pairs we observe a clear underspecification problem [45]: very similar ID performances (in most cases close to 1) lead to different OOD performances (vertical line). In this setting, ID accuracy is not a sufficient model selection criterion for obtaining robust models[3]. In some settings, the models do not transfer information from the ID to the OOD data at all and, despite having different ID performance, all models have very poor OOD performance. Finally, we observe a fourth setting, where OOD accuracy is hardly correlated to ID accuracy. Interestingly, we never see a decreasing trend, i.e., improved ID performance never systematically results into lower OOD performance. Hence, despite the many shapes of ID and OOD dependency, it is still a good strategy to maximize the ID accuracy in order to maximize the OOD accuracy.

**Results can significantly change for different shift types.** We highlighted how much ID to OOD generalization can change on different tasks/datasets. This is further confirmed by the task-specific correlation matrices in Appendices A.3 and A.4, which, more generally, show that there can be significant differences in various metrics between different tasks or shift types. For example, comparing the *terra-incognita* and *wilds-fmow* specific correlation matrices, we see that for *terra-incognita* calibration and demographic disparity have a strong *positive* correlation with OOD accuracy, whereas for *wilds-fmow* the correlation is strongly *negative*. Similarly, multi-domain calibration as proposed by [64] only improves OOD robustness on some tasks, but has a negative effect on others (details in Section 4.3). Appendix A.4 shows that focusing on different shift types can also lead to contradicting findings. For instance, for models that were trained on *artificial* data (such as sketches, clipart, simulated environments) and evaluated on *real* OOD data, corruption metrics are more predictive of OOD robustness than for models that were trained on *real* data and tested on *artificial* OOD data. Additionally, we discuss in Appendix E.1 the dependence of the results on the task difficulty.

---

[3]One may be tempted to think of this as a saturation phenomenon, where the ID data is too easy to learn to distinguish the good networks from the bad ones. In that case, however, the generalization properties should significantly depend on the architecture (and pre-training performance), so that models with best OOD performance should be the same on every dataset. What we observe instead is that the order seems to be largely random in different dataset pairs.

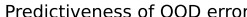

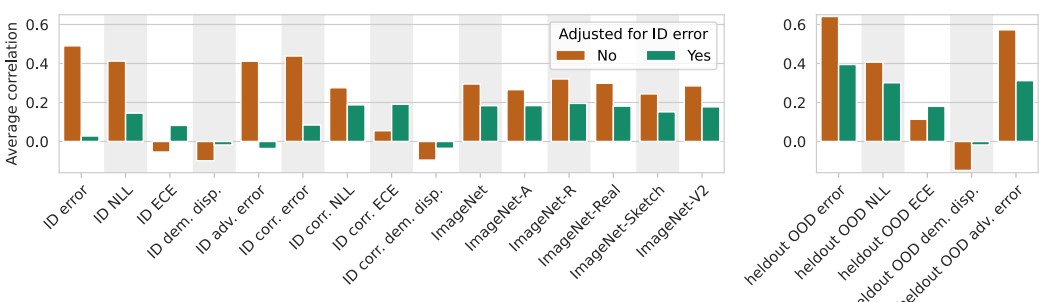

Figure 3: **LEFT:** *What is a good proxy for classification error under natural distribution shifts?* We measure how well several popular robustness metrics on in-distribution (ID) data predict classification error on out-of-distribution (OOD) datasets. **Red bars**: The predictiveness score is computed based on Spearman's rank correlation coefficient between the robustness metric and OOD classification error. We find that, among all considered metrics, ID classification error is the strongest predictor of OOD robustness. *What is the additional information content of the robustness metrics adjusted for ID classification error?* **Green bars**: We compute the adjusted predictiveness scores as outlined in Section 4.3. When adjusted for ID classification error, all secondary metrics only provide limited information. **RIGHT:** *How predictive are the metrics on additional held-out OOD data?* Evaluating accuracy on held-out OOD data (multi-domain evaluation) is the strongest predictor of OOD accuracy and provides significant additional information to ID accuracy (see adjusted scores).

> **Takeaway**: ID and OOD accuracy only show a linear trend on specific tasks. We observe three additional settings: underspecification (vertical line), no generalization (horizontal line), and random generalization (large point cloud). We did not observe any trade-off between accuracy and robustness, where more accurate models would overfit to "spurious features" that do not generalize. Robustness methods have to be tested in many different settings. Currently, there seems to be no single method that is superior in all OOD settings.

## 4.3 What are good proxies to measuring robustness to distribution shifts?

Can we predict the robustness of a model by using a proxy measure? In other words, how predictive is a certain *metric A* (e.g., ID expected calibration error) of another *metric B* (e.g., OOD classification error)? To this end we compute the *averaged correlation matrix* which reports the rank correlation of all metrics, averaged over all tasks. The matrix and details on the method are deferred to Appendix A. We already saw –and the matrix confirms– that accuracy is a strong predictor of OOD accuracy. This raises the question if other metrics add any additional information on OOD accuracy which is not already provided by ID accuracy. To test this, we compute adjusted predictiveness scores as follows. For each dataset pair, we fit a linear regression to predict OOD accuracy from ID accuracy. We then report the averaged rank correlation coefficient between the obtained residuals and each metric. This measure is similar to the *effective robustness* proposed in [35]. Results are shown in Fig. 3 and discussed in the upcoming subsections.

### 4.3.1 Overall classification error is the best general predictor of OOD robustness

Fig. S1, derived from the full averaged correlation matrix in Appendix A, shows that among all considered metrics, ID classification error is the strongest predictor of OOD classification error. This finding is in contrast to works that hypothesized that evaluating the classification error on corrupted data (e.g., ImageNet-C [17]) or on adversarial perturbed data [116] provides additional information on how models perform under natural distribution shifts. Although these metrics show a high correlation with OOD classification error, we do not find that they add significant information when adjusting for ID classification error. However, when having access to additional OOD datasets, the classification error on the held-out OOD datasets is even more powerful predictor of the robustness of the OOD dataset of interest, see Fig. 3 (right). We find that this is the most reliable model selection procedure of all considered metrics.

Our findings imply that if practitioners want to make the model more robust on OOD data, the main focus should be to improve the ID classification error. This is in accordance with previous work that

found that models with high ID classification error tend to be more robust [34, 35]. We speculate that the risk of "overfitting" large pre-trained models to the downstream test set is minimal, and it seems to be not a good strategy to, e.g., reduce the capacity of the model in the hope of better OOD generalization [38]. Finally, we recommend that architectural innovations and training techniques can leverage scale but that robustness comparisons should always be adjusted for classification error.

> **Takeaway**: Accuracy is the strongest ID predictor of OOD robustness and models that generalize well in distribution tend to also be more robust. Evaluating accuracy on additional held-out OOD data is an even stronger predictor.

### 4.3.2   What can we learn from other metrics beyond accuracy?

The first interesting result is that calibration on ID data is *not* predictive of OOD robustness or OOD log-likelihood (see Fig. S1 in the appendix). Restricted to the ID regime, however, we observe a correlation between ID calibration and ID classification error, which is in accordance with [65]. This difference is explained by the fact that ID calibration is not predictive of OOD calibration without an OOD held-out set (see Section 4.4). In contrast to the observations in [64], we see that a model that is well-calibrated on multiple domains (held-out OOD data) may not always have lower OOD classification error (e.g., negative correlation for domain-net but positive on office-home, see Appendix A.3). Interestingly, invariance measured with environment inference [77] and demographic disparity [103] is not predictive of OOD robustness but seems to be a good proxy for calibration of OOD data (see Fig. S1) which is consistent with our observations on multi-domain calibration[4] and may be useful for OOD detection.

ID log-likelihood and adversarial accuracy are both weak predictors of OOD robustness compared to ID accuracy, and when adjusted for ID accuracy they only add marginal to no information. Since the correlation between ID adversarial classification error and OOD classification error is fully explained by ID accuracy (see Fig. 3, left) suggests that adversarial distribution shifts do not characterize well natural distribution shifts.

**Synthetic corruptions** We apply the synthetic corruptions proposed by [17] to all datasets. First, we find that classification error and log-likelihood evaluated on the corrupted data are strongly correlated to OOD classification error (see Fig. 3, left). However, we find that the information provided by the corrupted metrics is significantly reduced when adjusted for ID accuracy. With the partial exception of corrupted calibration being more informative of OOD calibration than ID calibration (see Section 4.4). In summary, evaluation on corrupted data does not seem to bring the same benefits as using real held-out OOD data (see Fig. 3, right). Interestingly, we find that adversarial classification error is highly correlated to the classification error under synthetic corruptions (see Fig. S1). Therefore, if the practitioner cares about shifts defined by artificial corruptions, studying the adversarial robustness on ID data will be informative.

**Robustness to upstream dataset shifts** In our study all models are pre-trained on ImageNet (upstream dataset) and then fine-tuned on downstream data. In this section, we explore if upstream robustness propagates downstream. First, we notice in Fig. 3 (left) that the original performance on ImageNet is linked to OOD classification error in accordance to previous studies [34]. When we adjust for ID classification error, the clean ImageNet performance is among the strongest predictors for OOD classification error. Second, we find that robustness on ImageNet shifts does not give much additional information to the downstream robustness compared to clean performance. The performance on ImageNet shifts is almost perfectly correlated with the ID performance (in this setting accuracy is perfectly "on the line", c.f. Section 4.2), but this relationship does not translate to our diverse set of downstream shifts.

> **Takeaway**: Other metrics can add marginal additional information for OOD robustness. Calibration appears to be predictive of ID accuracy but does not transfer to new distributions and adversarial robustness appears not to reflect robustness to natural distribution shifts. Corruptions are only marginally useful for measuring robustness to natural distribution shifts and should not be used as a substitute to real held-out OOD data. ImageNet upstream performance provides information on downstream robustness. However, robustness to commonly used shifts of ImageNet does not imply downstream robustness more than the clean upstream accuracy.

---

[4]Given the decomposability of the log-score, the objectives of both approaches are related.

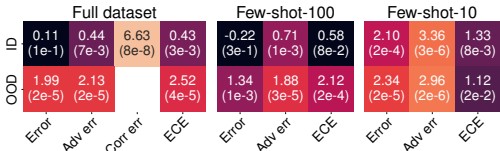

Figure 5: Performance gap (difference) between models trained with and without augmentations together with a p-value in parenthesis to assess its significance. Black fields indicate a p-value above the 0.05 significance threshold (i.e., non-significant); the other values are significant. Overall, augmentations help increasing the model's accuracy and its robustness to all kinds of distribution shifts (artificial and adversarial corruptions, OOD generalization), more so when data is scarce (few-shot settings).

| Model | ID Error | OOD Error | OOD-ID Gap |
|---|---|---|---|
| Deit | **0.101** ± 0.005 | **0.364** ± 0.008 | 0.263 ± 0.006 |
| Swin | 0.111 ± 0.005 | 0.371 ± 0.008 | **0.260** ± 0.006 |
| ViT-B | 0.124 ± 0.005 | 0.384 ± 0.008 | 0.259 ± 0.006 |
| ResNet50 | 0.124 ± 0.005 | 0.406 ± 0.008 | 0.283 ± 0.006 |
| EfficientNet2 | 0.129 ± 0.005 | 0.407 ± 0.008 | 0.277 ± 0.006 |
| GMLP | 0.140 ± 0.006 | 0.413 ± 0.008 | 0.273 ± 0.006 |
| ResMLP | 0.134 ± 0.005 | 0.413 ± 0.008 | 0.279 ± 0.006 |
| Mixer | 0.142 ± 0.006 | 0.425 ± 0.008 | 0.282 ± 0.006 |
| DenseNet169 | 0.145 ± 0.005 | 0.443 ± 0.008 | 0.298 ± 0.006 |

Table 2: Average classification error of model architectures with the standard error of this average in grey. To simulate a typical transfer learning workflow, we selected the best performing augmentations based on ID validation data for each fine tuning domain.

## 4.4 On the transfer of metrics from ID to OOD data

The main focus of Section 4.3 was to analyze how informative the different metrics are of OOD classification error. In a more general setting, we now explore how well a metric evaluated on ID data predicts their score on OOD data. Fig. 4 shows the averaged correlation coefficient of each metric—evaluated either on ID, corrupted or held-out data—with the same metric evaluated on OOD data. First, we find that all ID metrics transfer moderately well to OOD data (blue bars). For adversarial attacks the transfer is highest. This suggests that the models respond similarly to adversarial attacks on ID data and on OOD data. On the other hand, ID calibration transfers worst among all metrics, i.e., a model that is well calibrated on ID data, is not neces-

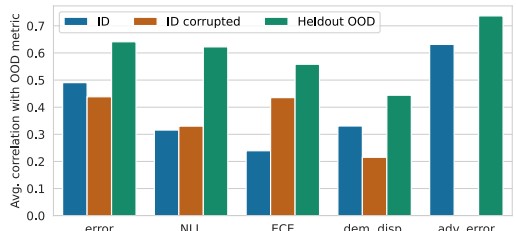

Figure 4: For each source metric on the x-axis we display the averaged correlation with the same metric evaluated on OOD data (target). The different colors indicate on which data domain the source metric was evaluated; either on ID, ID corrupted, or held-OOD data. Note that in our study we do not evaluate adversarial classification error on corrupted data.

sarily well calibrated on OOD data. This points to an important problem, since in many production systems models are only calibrated on ID data. Second, we observe that the evaluation on corrupted data does not add significant information to the evaluation on ID data (blue vs. red bars) for most metrics. Interestingly, we observe one exception; for calibration the evaluation on corruptions is significantly more informative. Third, when having access to additional held-out data, the evaluation on this data is the strongest predictor for the OOD behavior for all metrics (green bars).

> **Takeaway**: Among all metrics adversarial robustness transfers best from ID to OOD data, which suggests that models respond similarly to adversarial attacks on ID and OOD data. Calibration transfers worst, which means that models that are well calibrated on ID data are not necessarily well calibrated on OOD data.

## 5 The effect of the training strategy on out-of-distribution robustness

We now investigate the influence of the training strategy and model architecture on OOD robustness for practitioners. Although we will observe clear trends, **they should be taken with care**, since each model was *pre-trained* with its own training procedure (with different optimizers, learning rate schedules, augmentations, sometimes even datasets, etc.), which is likely to confound downstream results even after using a unified fine-tuning procedure. This is a general problem since different architectures usually require a specific pre-training procedure. Most practitioners usually undergo the same pipeline, starting from a network with publicly available pre-trained weights.

## 5.1 The effect of augmentations, fine-tuning strategy and few-shot learning

To evaluate the effect of augmentations during fine-tuning, we average the performance of networks trained with RandAugment [101] and AugMix [28] and compare it to fine-tuning without augmentations. Fig. 5 shows the performance gap between models trained with and without augmentations together with the p-value of a one-sided Wilcoxon signed-rank test that assesses whether the model trained without augmentations is better than the other one. Overall, augmentations appear to increase accuracy across all corruption types (natural, corrupted and adversarial data), particularly on OOD data. This suggests that augmentations not only improve accuracy in distribution, but also increase the model's robustness under certain shifts. The effect is more pronounced when data is scarce (few-shot setting), although exceptions exist (accuracy in "few-shot-100"). We discuss additional results in Appendix C.2.

Previous studies have shown that the fine-tuning strategy significantly affects the robustness [107, 117, 118]. In our study we investigate two popular fine-tuning methods: (1) fine-tuning the full architecture and (2) fine-tuning the head only, while keeping the rest of the architecture frozen. We discuss the results in Appendix C.1 and find that fine-tuning the full architecture is better for most of the considered tasks when having access to the full datasets. However, in the low data regime (few-shot-10 setting), fine-tuning the head only is beneficial on 40% of the tasks.

> **Takeaway**: Augmentations can improve accuracy and robustness to all kinds of distribution shifts (artificial and adversarial corruptions, OOD generalization), especially when data is scarce. While fine-tuning the full architecture is beneficial when having access to the full dataset, fine-tuning the head only can lead to higher robustness in the low data regime.

## 5.2 The effect of the model architecture

With many pre-trained backbones available in libraries like [100] that often achieve very similar results on ImageNet, it is not obvious whether the architecture choice matters. Table 2 shows the average ID and OOD classification errors of each model. Interestingly, we observe that while the Vision Transformer ViT-B was trained on more data it performs worse than Swin- and Deit-Transformers both on ID and OOD data (both approx. $3\%$ higher error than Deit). This indicates that the extensions made to vision transformers improve generalization performance in the transfer learning and fine-tuning scenario, while additionally requiring less data. Further, we notice that the model with lowest average OOD classification error, does *not* show the lowest performance gap, i.e., the performance on ID data and OOD data are not necessarily more closely aligned when performance on ID and OOD accuracy increases.

> **Takeaway**: In the light of previous work that argued that domain generalization methods only have a marginal effect on OOD robustness [75], we encourage more research on robust architectures, as our results indicate that the architecture can indeed make a difference.

## 6 Conclusions

In this paper, we thoroughly investigated out-of-distribution generalization and the interplay of several secondary metrics in the transfer learning setting. We focused on understanding sometimes contradicting empirical evidence from previous studies and on reconciling the results with anecdotal evidence from common practice in computer vision. We fine-tuned and evaluated over $31\text{k}$ models across several popular architectures on 172 (ID, OOD)-dataset pairs and found the following. **(1)** The risk of overfitting on the transfer distribution appears small: models that perform better in distribution tend to perform better OOD. All other proxy metrics convey only limited information on OOD performance after adjusting for ID accuracy. **(2)** Out-of-distribution generalization is a multi-faceted concept that cannot be reduced to a problem of "underspecification" [45] or to simple linear relations between ID and OOD accuracies [35]. However, we did not observe any trade-off between accuracy and robustness, as is commonly assumed in the domain generalization literature [73, 76–79]. While such trade-offs may exist, we posit that they may not be very common in non-adversarially chosen test sets. **(3)** While calibration appears to transfer poorly to new distributions, adversarial examples and synthetic corruptions transfer well to OOD data but seem ill-suited to mimic natural distribution

shifts. **(4)** Held-out OOD validation sets can be good proxies for OOD generalization. As such, they should be a key focus of any practitioner who worries about distribution shifts at test time.

In light of these results, we suggest three critical areas for further research. **(1)** Creating synthetic interventional distributions is an appealing alternative to hand-crafted augmentations and corruptions to both evaluate and improve robustness. High-fidelity generative models could be used to identify specific axes of variation that a model is not robust to. While this has been studied in the context of fairness with labelled sensitive attributes [e.g., 106], discovering such factors of variation remains an unsolved task that relates to disentanglement [119] and causal representation learning [46]. **(2)** While fine-grained studies of OOD performance can shed light into specific generalization properties of neural networks, they should be interpreted with care. In particular, conclusions from adversarially constructed test sets should not be generalized to broader settings. Instead, they may be useful to compile model cards [120] that contain specific strengths and weaknesses of a model, e.g., in terms of robustness to certain transformations, since we saw that these properties can transfer to new distributions. **(3)** More work is needed to understand whether inductive bias in the architecture is a meaningful tool to tackle generic distribution shifts. While we did observe some architecture-specific differences in performance, the many confounding factors during pre-training make it difficult to draw any definitive conclusion on this matter. Experimental protocols that specifically investigate the intrinsic robustness of architectures and its relation to ID accuracy are still required.

## Contributions

**Florian, Andrea, Peter, Carl-Johann, Max, Dominik, David and Francesco** contributed to the codebase.

**Max, David and Peter** designed and implemented the first version of the code.

**Andrea** initiated the first robustness experiments.

**Florian** conducted the main experiments and prepared the results for further analysis.

**Florian, Andrea, Carl-Johann, Max, Dominik, Chris and Francesco** contributed to the analysis and the interpretation of the results.

**Florian and Andrea** conducted the correlation analysis of robustness metrics.

**Carl-Johann and Chris** conducted the factor analysis.

**Carl-Johann** prepared the factor loadings and scatter plots, and analyzed the "many facets of out-of-distribution generalization".

**Max** analyzed the in-distribution vs. out-of-distribution performance gap.

**Dominik** analyzed the effect of augmentation and fine-tuning strategies on robustness.

**Francesco** proposed and advised the project.

**Thomas, Bernt and Bernhard** provided additional valuable insights and regular feedback.

**All authors** contributed to the writing of the paper.

**Florian** led the project.

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
