# A   Details on the averaged correlation matrices

## A.1   Method details

Our goal is to measure how predictive a *metric $A$* is of another *metric $B$*. We measure the predictiveness based on Spearman's rank correlation coefficient between all combinations of metrics on ID, OOD, corrupted and held-out OOD data. One issue with this measure however is that, while our empirical study covers multiple tasks with multiple (ID, OOD) dataset pairs, a correlation coefficient can only be computed for a specific dataset pair: pooling the data together from multiple dataset pairs (experiments) would lead to skewed results. For instance, for one dataset pair the ID and OOD datasets could be intrinsically hard, leading to low metric values, and for another dataset pair, the datasets could be intrinsically easy, leading to high values. Pooling the data of those two dataset pairs would lead to the impression of a clear (linear) trend which would not be necessarily present when considering the dataset pairs individually. Hence, trends should only be considered at the individual dataset level. Other studies circumvent this problem by reporting the correlation coefficients for each dataset pair individually (e.g., see [35, Table 1]), but since we have a much larger study with 172 dataset pairs, such a table would not be informative. Therefore, we report the *averaged correlation coefficient* of the 172 dataset pairs. We note that *this score cannot be interpreted as a statistical correlation* (due to the averaging) *but still serves as a meaningful measure of predictiveness*. For example, if we think in terms of Pearson (rather than Spearman) correlations, then an average (Pearson) correlation of 1 would mean that, on every dataset pair, we observe a linear dependency, but the parameters of this dependency (slope and intercept) could be different for every dataset pair. In particular, it would not mean that the pooled data lies on a single straight line.

The averaged correlation matrix is displayed in Fig. S1 and computed as follows. For a given (ID, OOD) dataset pair, a metric $A$, and a metric $B$, we first compute the Spearman's rank correlation coefficient [121] of the metrics on that dataset pair. To this end, we pool the data of all model architectures, augmentation and fine-tuning methods and compute the correlation of the corresponding metric scores. Second, to obtain an aggregated predictiveness score over all dataset pairs, we computed the *weighted average* of the correlation coefficients for all dataset pairs in each task. Since some tasks entail more dataset pairs then others, we compute a weighted average, such that each task has the same contribution to the final average predictiveness score. In summary, the averaged correlation coefficient between metric $A$ and metric $B$ is

$$\text{score}(A, B) = \sum_{i \in I} w_i \text{corr}\left(A|\text{ID}_i; B|\text{OOD}_i\right),$$

where $A|\text{ID}_i$ are the metric scores of metric $A$ evaluated on the dataset $\text{ID}_i$ and $B|\text{OOD}_i$ are the metric scores of metric $B$ evaluated on the dataset $\text{OOD}_i$. $I$ is the index set of all dataset pairs. The weight $w_i = 1/|T_i|$ normalizes by the number of dataset pairs in task $T_i$. See Appendix E on how the tasks are defined. In addition to the (ID, OOD) dataset pairs, we also compute the correlation of metrics on all other combinations of ID, OOD, and held-out OOD domains.

**Remark: Pearson correlation coefficients.**   Instead of computing rank correlation coefficients between metrics, we also tried using Pearson (linear) correlation coefficients. Qualitatively, this led to similar results as using rank correlations (Fig. S1). The order of the predictiveness of metrics stayed the same (e.g., ID classification error still had the highest correlation coefficient with OOD classification error). However, since some trends are not linear (see Appendix D) we found that rank correlation coefficients lead to more robust results.

**Adjusted predictiveness scores.**   As briefly described in Section 4.3, we assess the usefulness of a metric to explain OOD accuracy by the additional information they provide on top of what is already explained by ID accuracy. The adjustment is similarly done as in [110]. For each (ID, OOD) dataset pair, we first compute a linear regression between ID and OOD classification error. The residuals of that regression is the variance in the data that is not explained by ID classification error. Second, we compute the correlation coefficient of the metric of interest with the residuals, i.e., we compute how informative the metric is of the variance that is not already explained by ID accuracy. Again, we repeat this procedure for all (ID, OOD) dataset pairs and compute the weighted average as explained in Section 4.3. The adjusted predictiveness scores are displayed in Fig. 3 (green bars).

The adjusted predictiveness scores give a better sense of the usefulness of the metrics. For instance, inspecting Fig. S1, the averaged correlation between ID corrupted classification error and OOD

classification error is 0.44, which is only 0.05 points lower than the score for ID classification error. Only looking at this number, we would be tempted to conclude that the ID corrupted classification error is actually a good predictor of robustness and, hence, might be worth the additional compute burden. However, if we compute the adjusted predictiveness score we obtain a score of 0.01, see Fig. 3 (left, green bar). Hence, ID corrupted classification error provides no information that is not already covered by the standard ID classification error. This is also reflected by the fact that ID classification error and ID corrupted classification error are highly correlated (with an averaged correlation coefficient of 0.68, see Fig. S1).

## A.2 The full correlation matrix

We display the full averaged correlation matrix in Fig. S1. It shows the pair-wise averaged correlation coefficient between all combination of metrics and is computed as described in Appendix A.1. This matrix is the main source for our evaluations. For instance, Fig. 3 highlights one part of this matrix, focusing on the correlation of metrics with OOD classification error. The relevant row in the matrix is marked by a green rectangle.

## A.3 Analyzing each task separately

In Section 4.2 in the main text we argue that out-of-distribution generalization has many facets. Some patterns observed in prior work actually only hold on a subsets of the tasks included in our study. To investigate the stability of observations across different tasks we compute the averaged correlation matrices for each task separately. For each task, we only include the datasets that belong to that tasks (see Appendix E) and show the matrices in Figs. S2 and S3.

Some observations are stable across all tasks (e.g., classification error is always the strongest predictors on ID and held-out data, respectively). For other metrics we observe large variability across tasks. For instance for some tasks, demographic disparity has a strong positive link to OOD accuracy, for others it is reversed (i.e., here lower demographic disparity leads to a higher classification error). This also holds for calibration error. We marked the according cells in the task-specific correlation matrices with a green square. We also observe mixed positive and negative links for multi-domain calibration (held-out OOD ECE, marked by yellow squares in the plots). This is in contrast to the claims by [64] that multi-domain calibration is generally a good indicator for OOD robustness since it actually only holds on a subset of the tasks.

## A.4 Analyzing each shift type separately

In this section we investigate how results change for different shift types. We define the shift type based on ID dataset regime and the OOD dataset regime as follows. First, we label each dataset by one of the domain types *artificial* and *real*, see Table S1. For instances, the label *artificial* refers to datasets of images composed from sketches, clipart or simulated environments. Second, for an (ID, OOD) dataset pair the shift is derived by the ID (source) domain type and the OOD (target) domain type. E.g., the dataset pair: (OfficeHome-ClipArt, OfficeHome-RealWorld) is labeled by the shift type *artificial* → *real*.

Similarly, as in Appendix A.3, we compute the averaged correlation matrix for the subset of dataset pairs corresponding to each shift type separately. The matrices are displayed in the figures Fig. S4. For instance, for the shift type *artificial* → *real*, corruption metrics are more predictive of OOD accuracy then for the reversed shift type *real* → *artificial*, see the highlighted green cells in the plots.

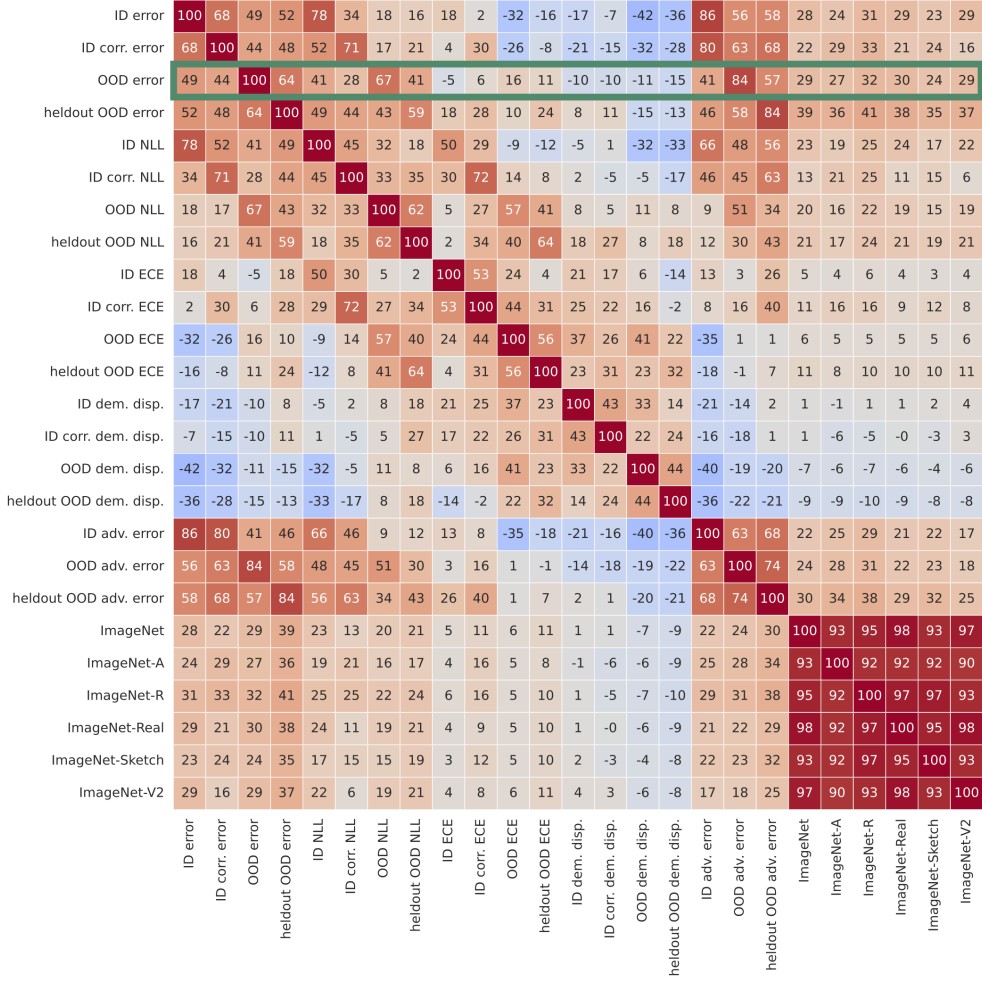

Figure S1: Rank correlation matrix averaged over all tasks. The matrix shows the predictiveness of a metric (either evaluated on ID data, ID corrupted data, OOD, held-out OOD data or ImageNet) of another metric. Positive values indicate a positive link between the metrics, i.e. better results for one metric tend to lead to better results for the other metric. Negative values indicate a negative link, i.e. better results for one metric tend to lead to worse results for the other metric. All averaged correlation coefficients are multiplied by a factor of 100 for better readability. As an example, the highlighted **green rectangle** shows how well metrics predict OOD classification error, this information is shown separately as a bar plot in Fig. 3 (red bars) in the main text.

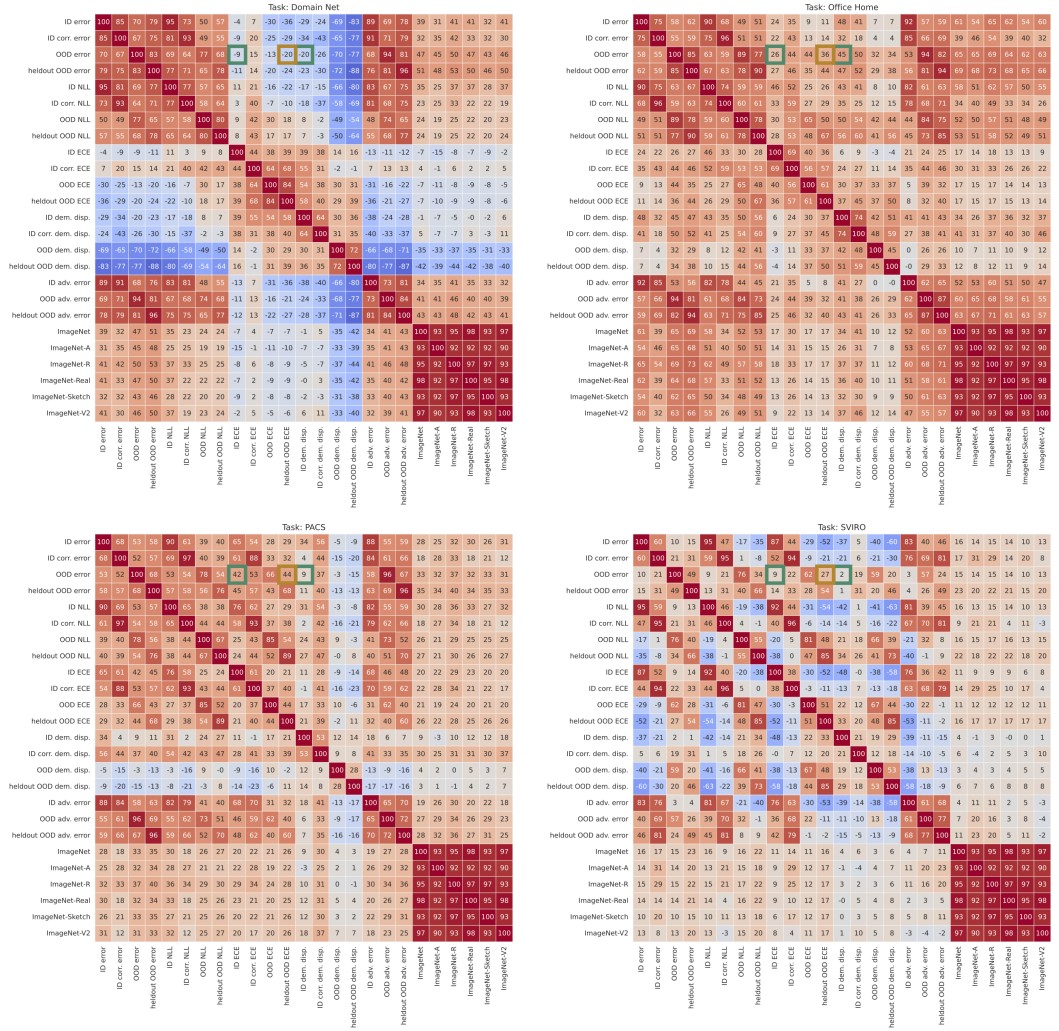

Figure S2: Averaged correlation matrices for each task separately. For each task we only consider the corresponding datasets and compute the correlation coefficients separately. We highlight differences between tasks for ID demographic disparity, ID calibration with **green squares** and differences for held-out ECE with **yellow squares**.

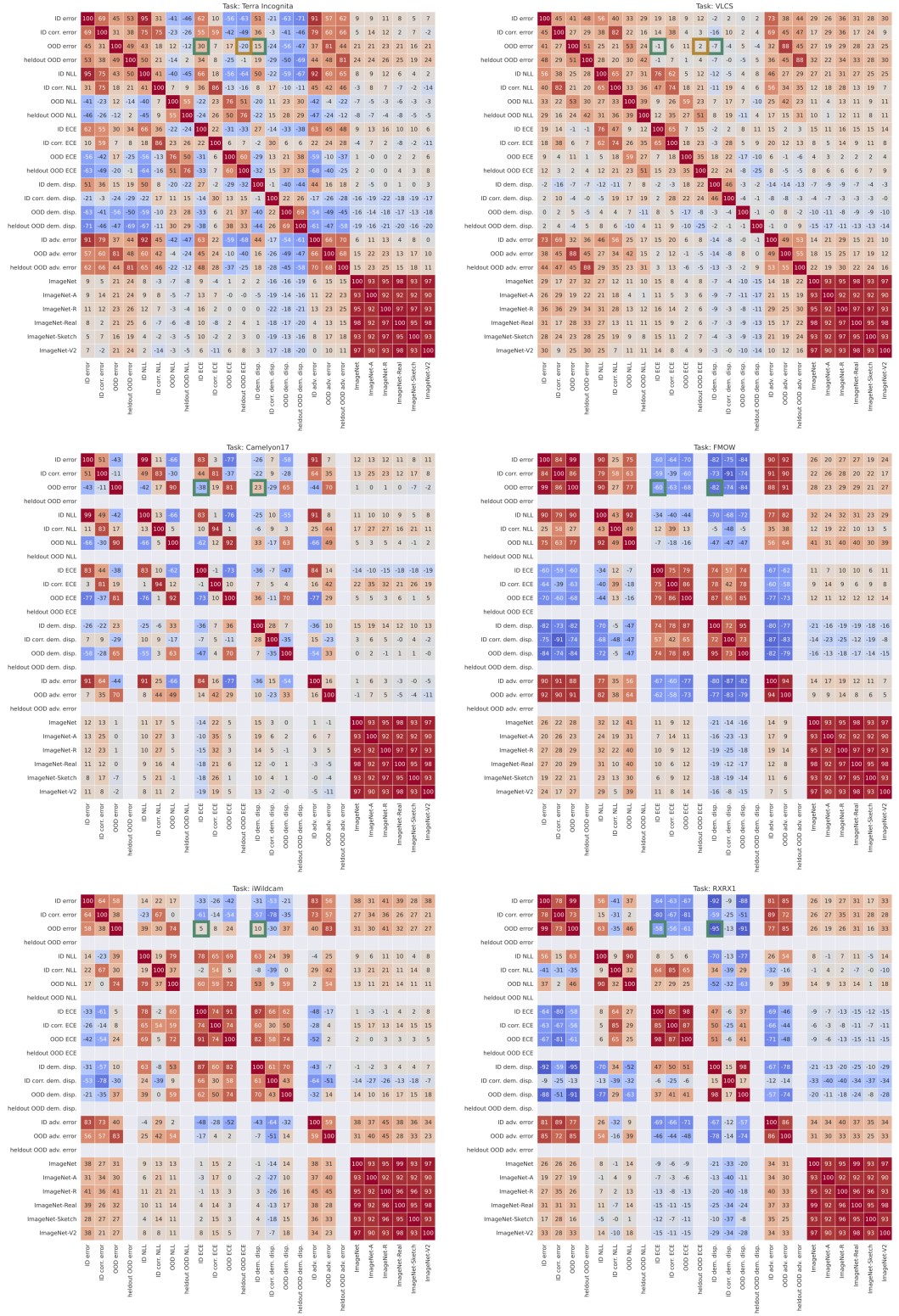

Figure S3: Averaged correlation matrices for each task separately (continued). We highlight differences between tasks for ID demographic disparity, ID calibration with **green squares** and differences for held-out ECE with **yellow squares**. Note that for the tasks *Camelyon17*, *FMOW*, *iWildcam* and *RXRX1*, no held-out OOD data is available (i.e., only one OOD split is provided).

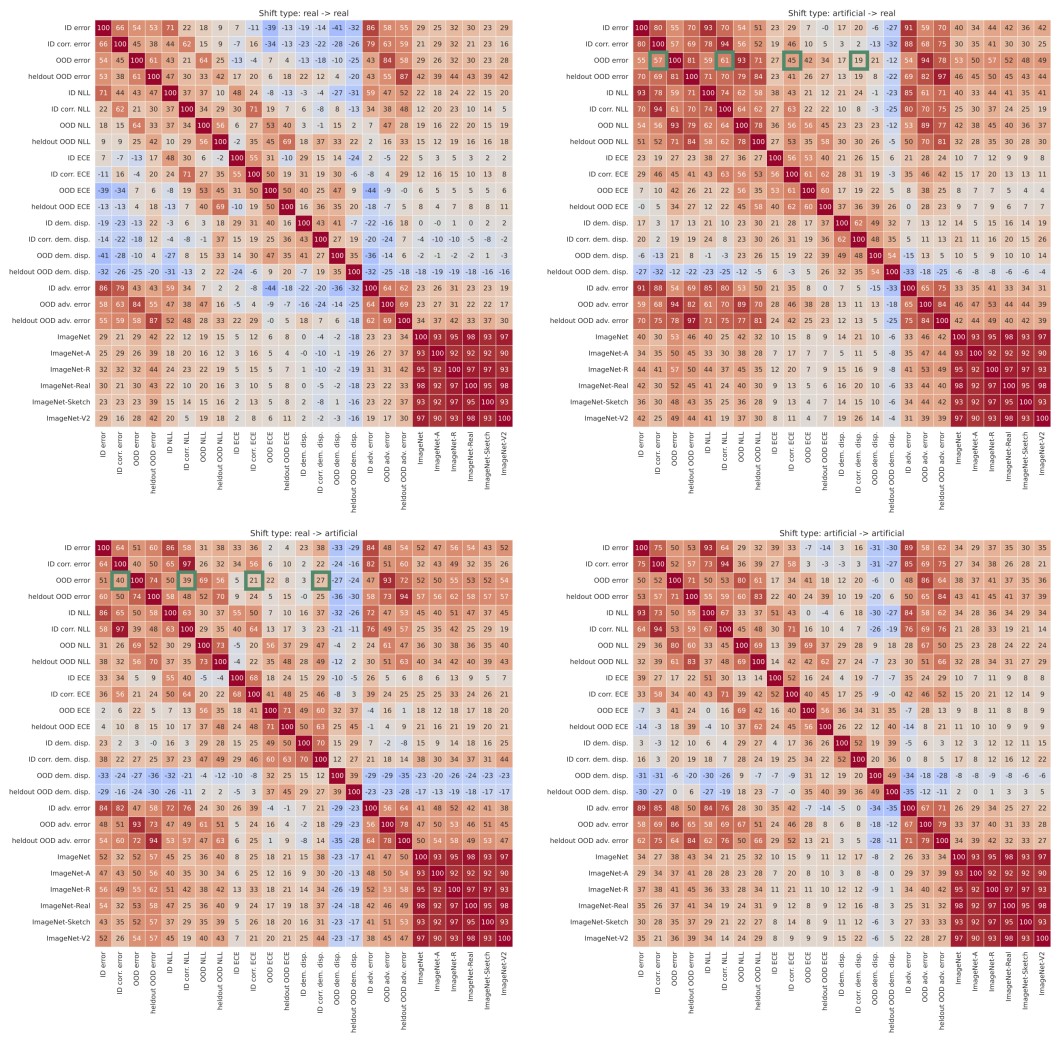

Figure S4: Averaged correlation matrices for each shift type separately. The shift types are defined based on the source and target domain, see Table S1. Using **green squares** we highlight differences of the corruption metrics predicting OOD accuracy for the shift types *real → artificial* and *artificial → real*.

# B Factor analysis: details and background

This section gives a short introduction to factor analysis, provides details about the data and the pre-processing used for the factor analysis of Section 4.1 and presents the data's scree plot in Fig. S5.

**Background.** Factor analysis is a well-established tool from statistics used to analyze tabular data, such as data arising from a poll, where the columns would be the questions and the rows the answers of each individual. It aims at finding a minimal amount of latent variables called *factors* that suffice to explain the data and reflect the main dependencies between columns. In particular, given $k$ factors $f_1, f_2, \ldots, f_k$, each column $c_i$ gets approximated as $c_i = \lambda_{i1} f_1 + \cdots \lambda_{ik} f_k$ where the $\lambda_{ij}$s are called factor *loadings*. Because columns get standardized prior to the analysis and factors are scaled to have unit norm, the loadings lie between $-1$ and $1$. While the factors are a priori abstract variables with no pre-defined interpretation, one of the main challenges and goals of factor analysis is to understand what variability they capture by analyzing their loadings. Note that the first $k$ factors span the subspace defined by the first $k$ eigenvectors of the data matrix. However, the factors themselves are only defined up to a rotation, since rotating the factors and the loadings accordingly, the $c_i$s remain unchanged. Some rotations lead to factors whose loadings are easier to interpret than others. Therefore, the typical workflow of a factor analysis is as follows. (1) Decompose the data into eigenvectors and use the eigenvalues to choose the number $k$ of factors to keep. The simplest rule of thumb is to keep all eigenvectors with eigenvalues $\geq 1$. (2) Try out different rotations of the $k$ first eigenvectors, to make the loadings as easy as possible to explain or interpret. To do so, one typically just tests several standard rotation methods such as the *varimax*, *quartimax*, or *equamax* methods (which typically try to promote some form of sparse loadings). (3) Interpret the obtained factors and conclude on the relations between the columns that they show.

**Data preprocessing.** We organize our data in a table with rows being fine-tuned networks and columns the metrics. Specifically, each network is defined by its model architecture, the adaptation dataset used for fine-tuning and the training parameters (augmentation strategy, learning rate, number of training epochs, and fine-tuning method), leading in theory to a total of 7776 networks (9 architectures $\times$ 36 datasets $\times$ $3 \cdot 2 \cdot 2 \cdot 2$ parameters). Note that we considered only those networks finetuned on full datasets, ignoring the few-shot data. For each network, we considered a total of 16 metrics: the 6 base metrics (cf. Section 2) computed respectively on the held-out test data from the adaptation domain (the in-distribution metrics), on its corrupted variant (except adversarial accuracy, which we did not compute on corrupted data), and, for each metric, its *average* over all out-of-distribution datasets from the adaptation dataset's task. Concerning the log-likelihood metric, most values lied between $0$ and $5$, but some outliers could take values up to $1\mathrm{e}10$. We therefore mapped all log-likelihood values through the following function: $f(x) = 10(1 - \exp(-x/10))$. Doing so ensures that all typical log-likelihood values remain nearly unchanged up to a rescaling factor ($f$ is almost linear in $0$), while bigger values saturate at $10$. We used python's `factor_analyzer` package for the analysis.

**Factor analysis and factor loading plot.** The scree plot in Fig. S5 shows the eigenvalues of the data in decreasing order. We decided to retain $4$ factors, since only $4$ eigenvalues were $\geq 1$. We then used the *equamax* rotation method, but the results do not change significantly with other standard methods. The factor loadings for each metric are shown in Fig. 1 in the main text and discussed there (see Section 4.1).

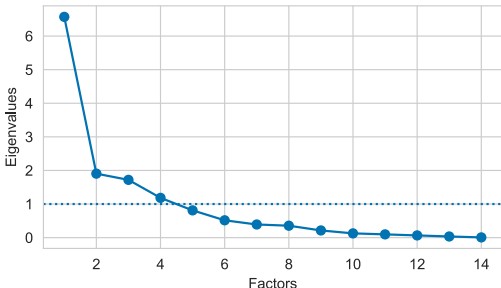

Figure S5: *Scree plot*: Eigenvalues of the standardized data from the factor analysis, sorted by decreasing order of magnitude. As explained in Section 4.1, we decided to take the first 4 factors, since only the first 4 eigenvalues are above the usual threshold value of 1.

## C   Detailed analysis on the effect of the fine-tuning and augmentation strategies

In the following we provide additional studies on the effect of the augmentation and fine-tuning method on the ID and OOD performance. The main results are summarized in the main text in Section 5.

### C.1   The effect of the fine-tuning strategy

Appendix C.1 shows the average ID and OOD accuracy for the two considered fine-tuning strategies: fine-tuning the full architecture and the linear probe classifier (fine-tuning the head only). To make the difference of both strategies more clear we show the normalized accuracy gap of both fine-tuning approaches in Appendix C.1. The gap is computed by

$$\text{gap} = \frac{\text{acc}_{\text{full}} - \text{acc}_{\text{head only}}}{\text{acc}_{\text{full}}}, \tag{S1}$$

where the accuracy terms are averaged over all datasets within a task. We find that fine-tuning the full architecture is usually superior when using the full fine-tuning dataset. Interestingly, when having access to less data (especially in the few-shot-10 setting), we observe that the linear probe classifier can be better, especially when evaluating on OOD data. This may confirm the idea that changing the last layer only leads to a higher inductive bias by the pre-training data. This is particularly beneficial in low-data regimes when generalizing to OOD data. This insights are in line with previous work (e.g., [122], Fig. 4).

### C.2   The effect of the augmentation strategy

In addition to Fig. 5 in the main text, we report the average ID and OOD accuracy for each augmentation method for each tasks separately. We do not observe a significant difference between the tasks.

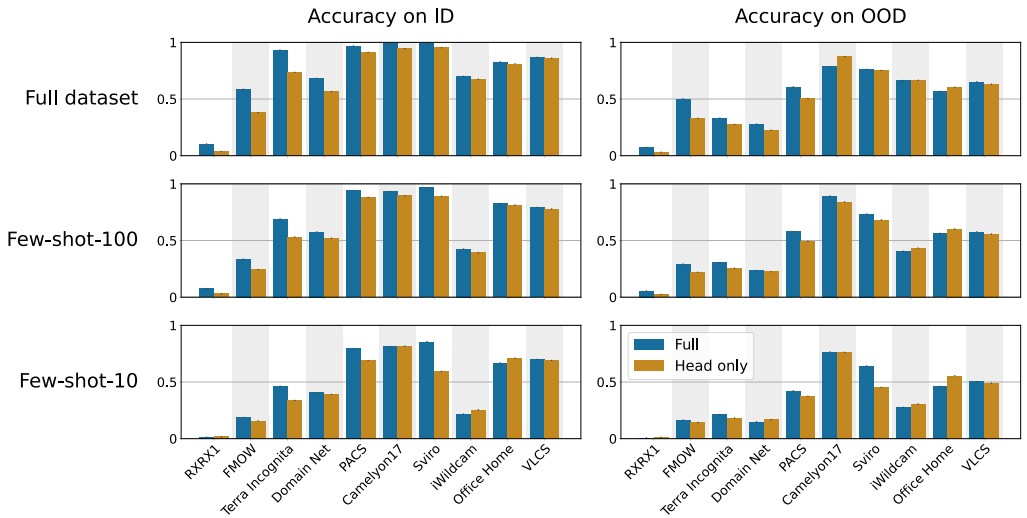

Figure S6: Each bar plot shows the average ID (left) and OOD accuracy (right) obtained by fine-tuning the full architecture or the head only. We compare the results in the setting of training on the full dataset and the few-shot settings (columns). In all plots, the black bars (which are very small and hence barely visible) indicate the standard error.

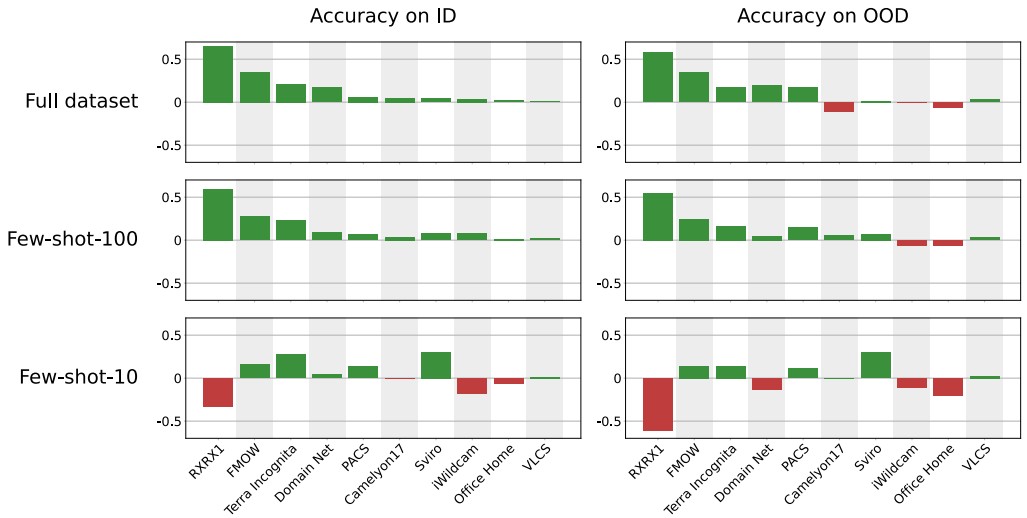

Figure S7: Each bar plot shows the normalized accuracy gap between fine-tuning the full architecture or the head only. We report the average ID (left) and OOD (right) accuracy in setting of training on the full dataset and the few-shot settings (columns). In the low data regime fine-tuning the head only can be beneficial, especially for OOD accuracy.

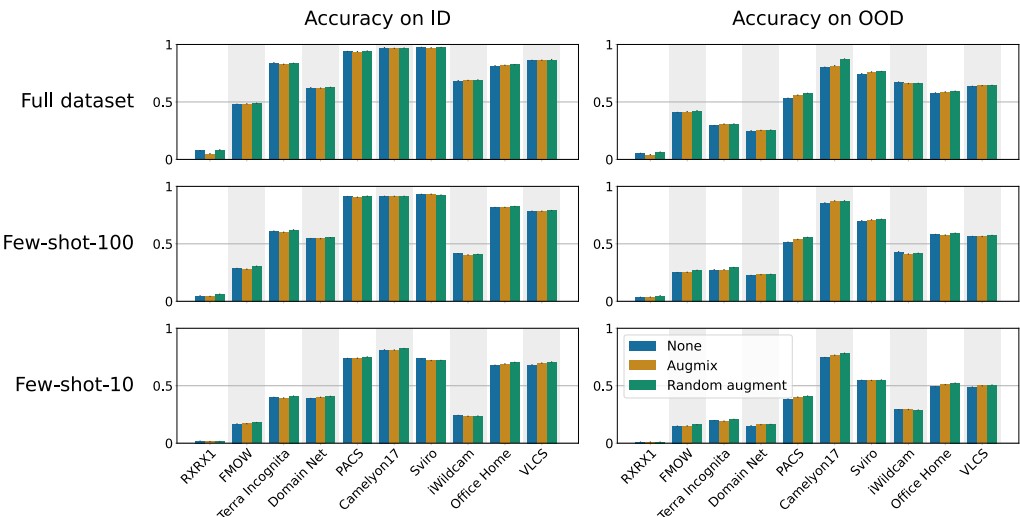

Figure S8: Each bar plot shows the average ID (left) and OOD accuracy (right) obtained by using the different augmentation strategies. We compare the results in the setting of training on the full dataset and the few-shot settings (columns). In all plots, the black bars (which are very small and hence barely visible) indicate the standard error.

# D  Detailed scatter plots for the relationship of ID vs. OOD accuracy

Figure 2 in the main text shows four prototypical patterns of ID versus OOD accuracy plots, taken from four exemplary dataset pairs out of the 172 dataset pairs considered in this study. This section now shows all 172 ID-vs-OOD-accuracy scatter plots in Fig. S9, grouped by task. Each column corresponds to one adaptation (i.e., fine-tuning) dataset. Interestingly, the exemplary scatter plot patterns described in Fig. 2 (functional relationship, vertical line/underspecification, no generalization/ horizontal line, and random generalization/point-cloud) appear to be essentially task related. For example, in OfficeHome and DomainNet, almost all (ID, OOD) dataset pairs exhibit a clear functional relationship. In Terra Incognita, generalization never works (horizontal line) or is near-random (unstructured point cloud). The SVIRO dataset pairs almost systematically fall into the underspecification/vertical line category, and PACS and VLCS exhibit either a relatively clear functional relationships or an underspecification pattern (vertical line).

For better readability, the plots do not include all degrees of freedom of our study. Specifically, we plot only those networks that were trained on the full adaptation dataset (i.e., we do not include the few-shot settings) and focus on the fine-tuning strategy where all weights of the network get fine-tuned (in contrast to fine-tuning the head only).

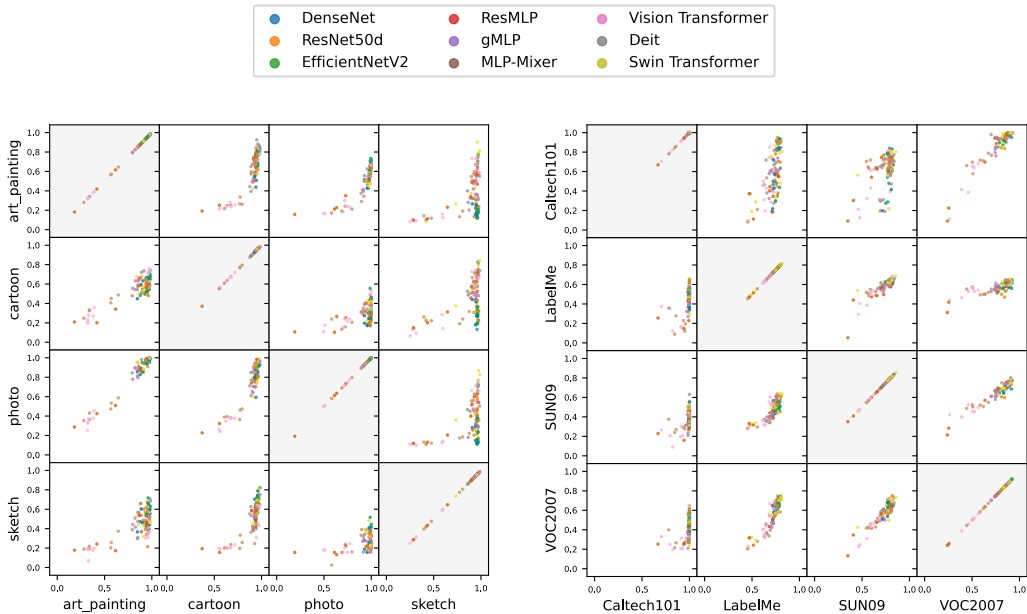

(a) **PACS**: We observe either a clear functional relationships or an "underspecified" regime (vertical line).

(b) **VLCS**: We observe similar patterns as for PACS.

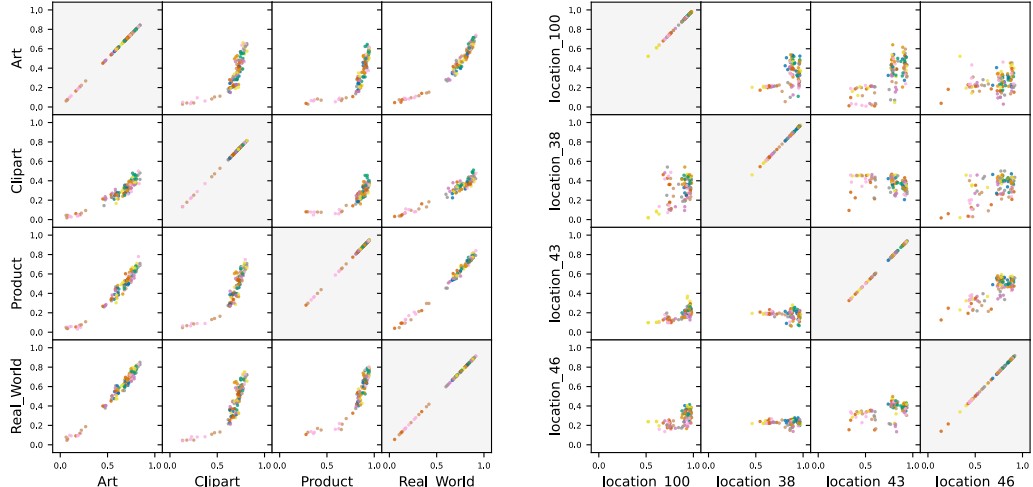

(c) **Office Home**: For all dataset pairs we observe clear functional relationships.

(d) **Terra Incognita**: Either no transfer (horizontal line) or random associations (scattered point cloud).

Figure S9: ID versus OOD accuracy for various tasks and dataset pairs. Every point represents one fine-tuned network. The domains on the x-axis are the ones used for fine-tuning and measuring the ID test accuracy. The datasets on the y-axis are the OOD datasets. We plot only networks that were trained on the full dataset (no few-shot datasets) and with all their weights (no head-only fine-tuning). All points in a same column with the same x-axis value represent the same fine-tuned network. Interestingly, the different patterns described in Fig. 2 (clear functional relationship, underspecification / vertical line, no generalization / horizontal line, random generalization / point-cloud) appear to be essentially task dependent. The reader may want to focus on the regions with higher point densities, since the networks outside those denser regions typically correspond to a suboptimal combination of hyperparameters that did not allow to reach convergence.

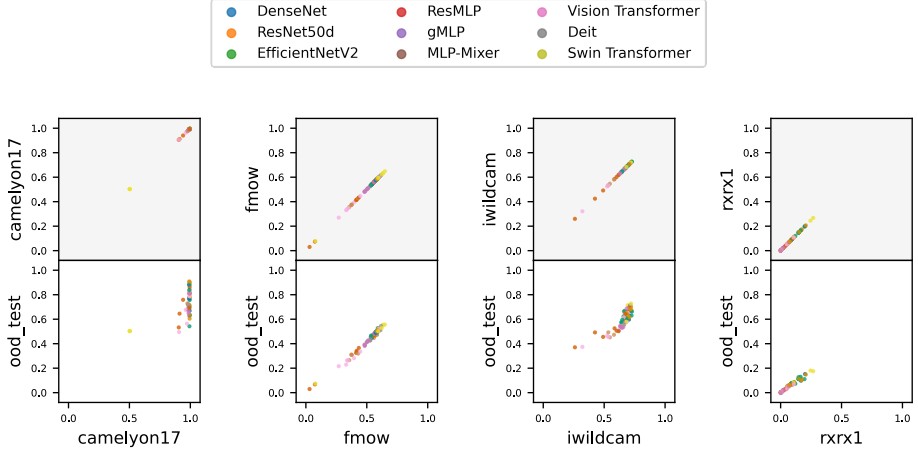

(e) **WILDS Camelyon17, FMoW, iWildCam, RXRX1**: We observe underspecification (Camelyon17) or clear functional relationships (others).

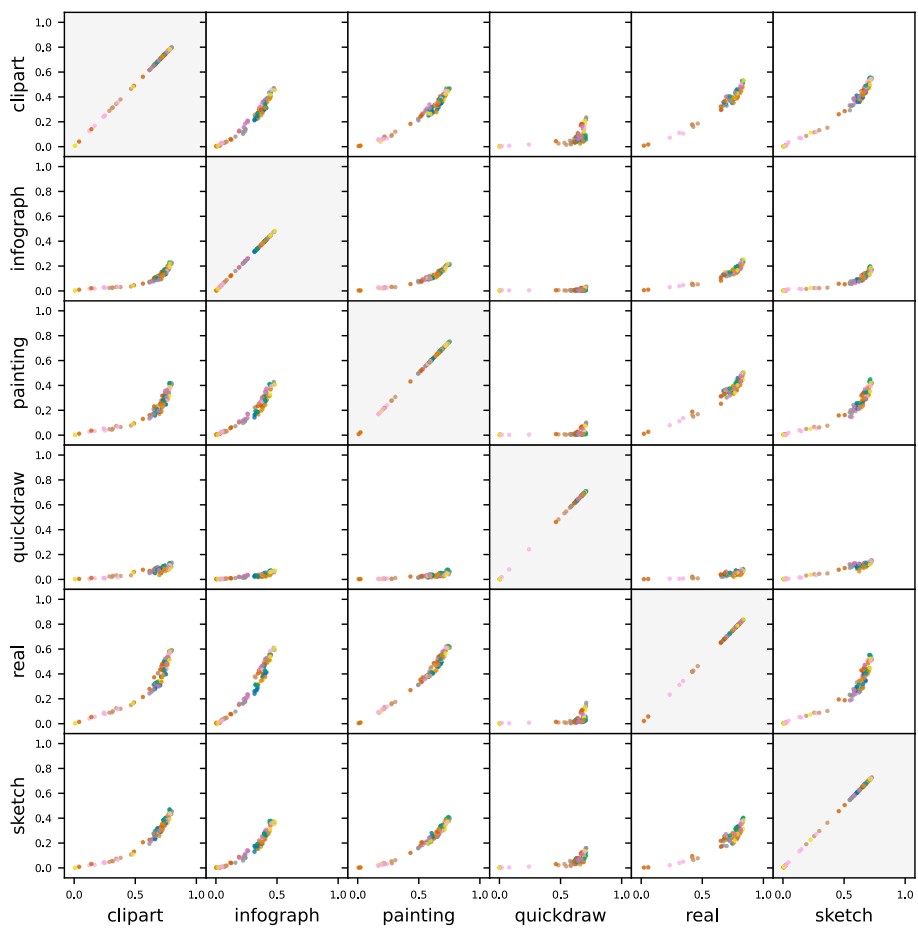

(f) **Domain Net**: For all dataset pairs we observe clear functional relationships.

Figure S9: (Continued) ID vs OOD accuracy for various tasks and dataset pairs. For a longer caption, see previous page.

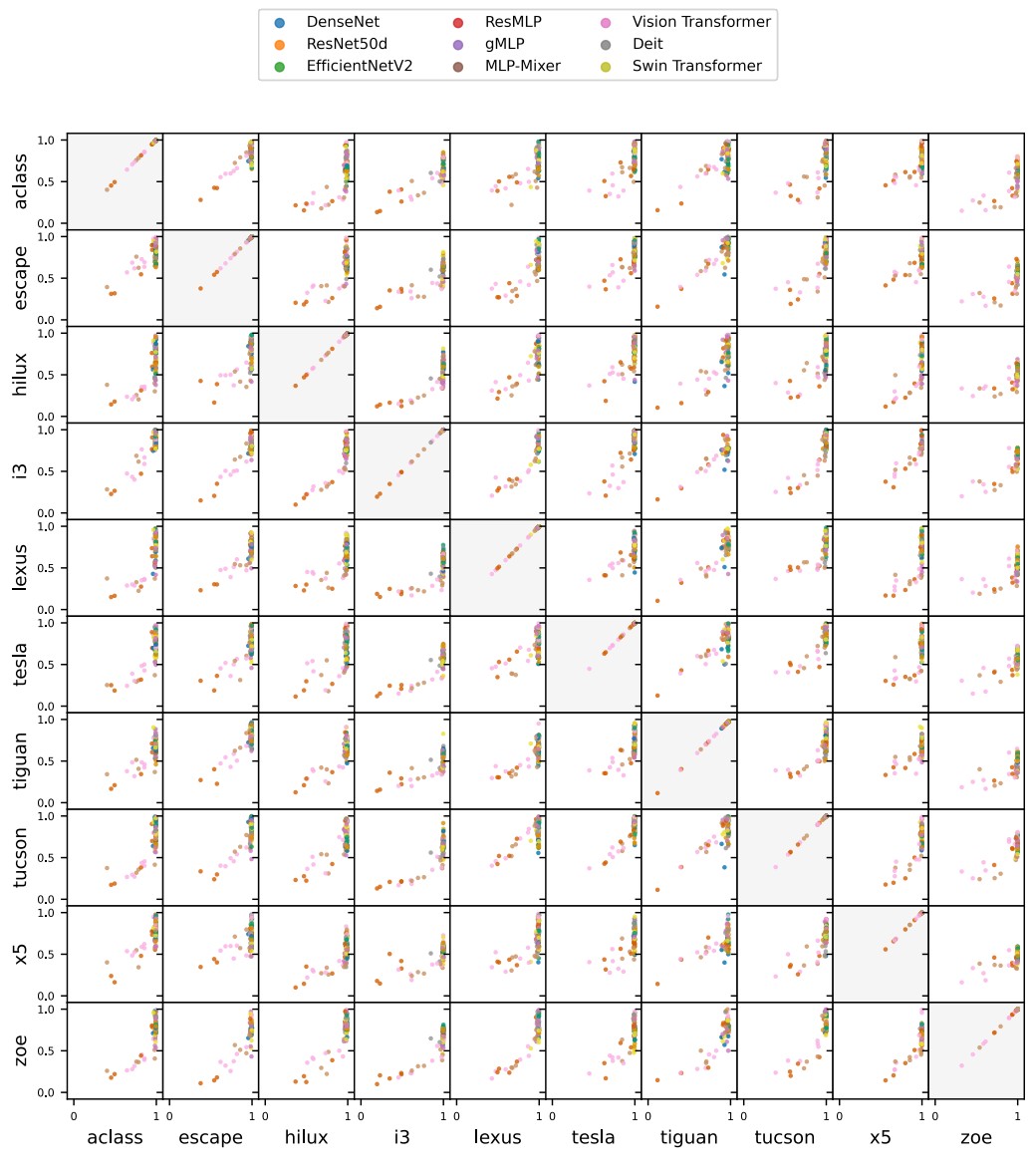

(g) **SVIRO**: almost always underspecified regime (vertical line).

Figure S9: (Continued) ID vs OOD accuracy for various tasks and dataset pairs. For a longer caption, see previous page.

# E   Details on the tasks and datasets

We use 36 datasets collected from different benchmarks: DomainNet [83], PACS [84], SVIRO [85], Terra Incognita [13] as well as the Caltech101 [86], VLCS [87], Sun09 [88], VOC2007 [89] and the Wilds datasets [90]. We group the domains (datasets) into ten different tasks, see Table S1 list. For all tasks besides the wilds tasks, we consider all possible (ID, OOD) dataset pairs, i.e., we fine-tune on one dataset in the task (ID dataset) and evaluate the model on all the others (which are considered to be the OOD datasets). For the Wilds datasets we use the predefined ID and OOD splits.

Table S1: Overview over tasks and their associated domains (datasets), along with a domain type tag.

| Task | Domain | Domain type |
|---|---|---|
| PACS | art_painting | Artificial |
| | cartoon | Artificial |
| | photo | Real |
| | sketch | Artificial |
| VLCS | Caltech101 | Real |
| | LabelMe | Real |
| | SUN09 | Real |
| | VOC2007 | Real |
| domain_net | clipart | Artificial |
| | infograph | Artificial |
| | painting | Artificial |
| | quickdraw | Artificial |
| | real | Real |
| | sketch | Artificial |
| office_home | Art | Artificial |
| | Clipart | Artificial |
| | Product | Real |
| | Real_World | Real |
| sviro | aclass | Artificial |
| | escape | Artificial |
| | hilux | Artificial |
| | i3 | Artificial |
| | lexus | Artificial |
| | tesla | Artificial |
| | tiguan | Artificial |
| | tucson | Artificial |
| | x5 | Artificial |
| | zoe | Artificial |
| terra_incognita | location_100 | Real |
| | location_38 | Real |
| | location_43 | Real |
| | location_46 | Real |
| wilds-camelyon17 | camelyon17-id | Real |
| | camelyon17-ood | Real |
| wilds-fmow | fmow-id | Real |
| | fmow-ood | Real |
| wilds-iwildcam | iwildcam-id | Real |
| | iwildcam-ood | Real |
| wilds-rxrx1 | rxrx1-id | Real |
| | rxrx1-ood | Real |

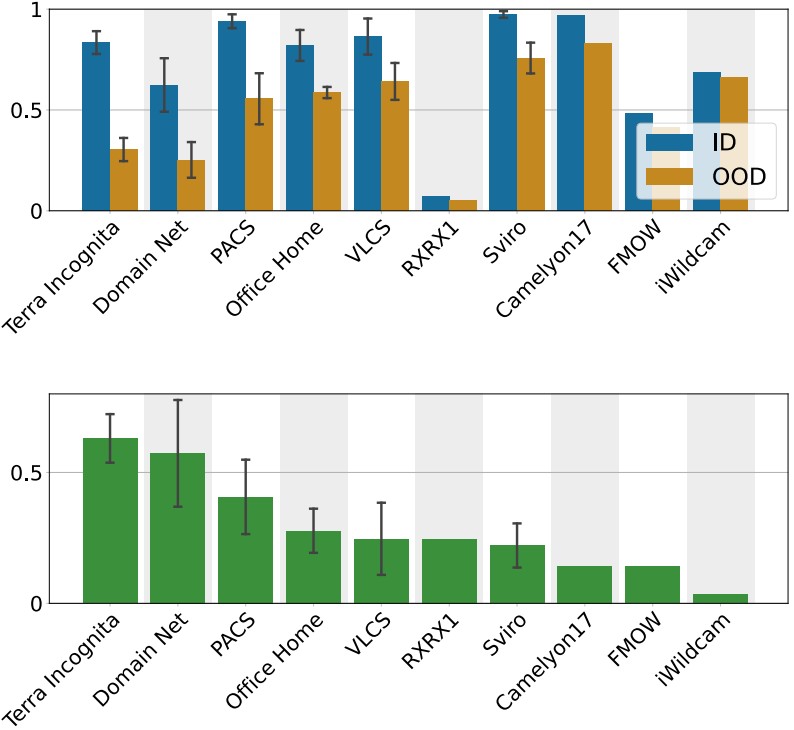

Figure S10: **TOP:** Mean accuracy on ID data and OOD data for each task (averaged over all datasets in the task, architectures, augmentation and fine-tuning strategies, c.f. Table S1). **BOTTOM:** Normalized mean ID vs. OOD accuracy gap for each task, see Eq. (S2). This indicates the average difficulty of the domain generalization. In all plots, the black bars indicate the standard deviation (not the standard error as in the other plots) across the datasets within the task if the tasks contains multiple dataset pairs.

## E.1 Comparison of the task difficulty

In the following we discuss the difficulty of the different tasks. In our context, the difficulty of a task might be defined by how hard it is to transfer from the ID dataset to the OOD dataset. In other words, a task is more difficult if the distribution of the OOD data is more distant from the distribution of the ID data. It is not obvious how to measure this distance and multiple approaches have been proposed [e.g., 12, 123]. Here we choose a more direct measure and compare the ID and OOD performance for each task averaged over all dataset pairs in the task, and all models and training hyperparameters. Fig. S10 shows the average accuracy obtained on all datasets within each task. In the top plot we compare the average ID accuracy and OOD accuracy. In the bottom plot we display the normalized ID vs. OOD accuracy gap computed by

$$\text{gap} = \frac{\text{acc}_{\text{ID}} - \text{acc}_{\text{OOD}}}{\text{acc}_{\text{ID}}}, \tag{S2}$$

where the accuracy terms are averaged over all datasets within a task. The average ID vs. OOD accuracy gap can serve as a proxy measure of the difficulty of a task. It indicates how hard the OOD prediction task is in average for a model that was trained on the ID data.

Next we discuss how the predictiveness of the different metrics depend on the task difficulty. Fig. S11 shows the correlation coefficients (between the ID metrics and OOD prediction error) as a function of task difficulty. We observe that ID error is the best predictor of the ID error for most difficulty levels. Besides this observation we find that the results are noisy and no clear pattern can be observed. Surprisingly, the correlation coefficients seem to be rather independent of the task difficulty.

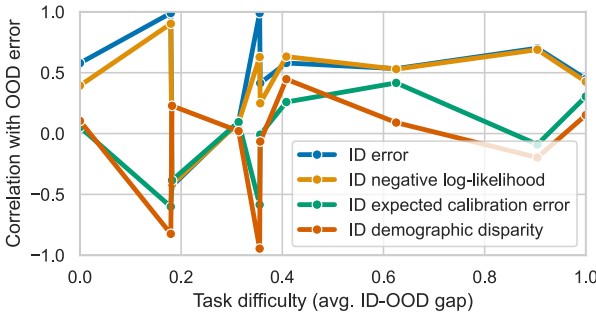

Figure S11: Each line shows the correlation coefficients between an ID metric and the OOD prediction error as a function of task difficulty. The task difficulty was computed by the normalized ID vs. OOD gap as shown in Fig. S10 and was linearly mapped to values between 0 and 1. No clear pattern is visible and the correlation coefficients seem to be rather independent of the task difficulty.

## F    Detailed comparison of model performances

We compare the in-distribution (ID) and out-of-distribution (OOD) performance for each model on all tasks. Here we follow the scenario where the hyperparameters are selected on an held-out ID validation split (this mimics the usual hyperparameter selection approach used in practice). We then compute the ID performance on the ID test set and the mean OOD performance on all other (OOD) domains of a task. Finally, we report the mean ID and OOD accuracy within each task and the gap between ID and OOD performance in Table S2. The numbers in gray represent the standard error of these means. This table represents a more details view on the performance of each individual model compared to Table 2 in the main text, where the performance is averaged over tasks.

Table S2: Average ID and OOD accuracy of each model on each task over all domains in the task.

| Model | Domain | ID Error | OOD Error | OOD-ID Gap |
|---|---|---|---|---|
| Deit | PACS | $0.033 \pm 0.003$ | $0.333 \pm 0.022$ | $0.299 \pm 0.023$ |
| | VLCS | $0.118 \pm 0.010$ | $0.328 \pm 0.017$ | $0.210 \pm 0.022$ |
| | domain_net | $0.316 \pm 0.009$ | $0.703 \pm 0.014$ | $0.387 \pm 0.017$ |
| | office_home | $0.131 \pm 0.008$ | $0.341 \pm 0.017$ | $0.209 \pm 0.020$ |
| | sviro | $0.014 \pm 0.001$ | $0.219 \pm 0.005$ | $0.205 \pm 0.005$ |
| | terra_incognita | $0.127 \pm 0.011$ | $0.661 \pm 0.013$ | $0.533 \pm 0.020$ |
| | wilds-camelyon17 | $0.022 \pm 0.007$ | $0.156 \pm 0.036$ | $0.134 \pm 0.040$ |
| | wilds-fmow | $0.468 \pm 0.036$ | $0.543 \pm 0.030$ | $0.074 \pm 0.007$ |
| | wilds-iwildcam | $0.275 \pm 0.007$ | $0.289 \pm 0.009$ | $0.014 \pm 0.006$ |
| | wilds-rxrx1 | $0.925 \pm 0.005$ | $0.939 \pm 0.004$ | $0.014 \pm 0.002$ |
| DenseNet169 | PACS | $0.061 \pm 0.006$ | $0.523 \pm 0.031$ | $0.463 \pm 0.031$ |
| | VLCS | $0.137 \pm 0.011$ | $0.365 \pm 0.018$ | $0.228 \pm 0.024$ |
| | domain_net | $0.387 \pm 0.011$ | $0.788 \pm 0.012$ | $0.401 \pm 0.015$ |
| | office_home | $0.209 \pm 0.010$ | $0.496 \pm 0.017$ | $0.287 \pm 0.019$ |
| | sviro | $0.047 \pm 0.003$ | $0.274 \pm 0.005$ | $0.227 \pm 0.006$ |
| | terra_incognita | $0.186 \pm 0.015$ | $0.725 \pm 0.015$ | $0.539 \pm 0.019$ |
| | wilds-camelyon17 | $0.036 \pm 0.012$ | $0.169 \pm 0.038$ | $0.133 \pm 0.047$ |
| | wilds-fmow | $0.526 \pm 0.046$ | $0.593 \pm 0.037$ | $0.067 \pm 0.009$ |
| | wilds-iwildcam | $0.319 \pm 0.010$ | $0.364 \pm 0.012$ | $0.046 \pm 0.021$ |
| | wilds-rxrx1 | $0.911 \pm 0.039$ | $0.937 \pm 0.027$ | $0.026 \pm 0.014$ |
| EfficientNet2 | PACS | $0.053 \pm 0.005$ | $0.469 \pm 0.032$ | $0.416 \pm 0.032$ |
| | VLCS | $0.140 \pm 0.012$ | $0.374 \pm 0.018$ | $0.234 \pm 0.023$ |
| | domain_net | $0.369 \pm 0.012$ | $0.734 \pm 0.014$ | $0.365 \pm 0.017$ |
| | office_home | $0.172 \pm 0.009$ | $0.403 \pm 0.015$ | $0.230 \pm 0.017$ |
| | sviro | $0.031 \pm 0.002$ | $0.249 \pm 0.005$ | $0.218 \pm 0.005$ |
| | terra_incognita | $0.187 \pm 0.016$ | $0.703 \pm 0.013$ | $0.516 \pm 0.017$ |
| | wilds-camelyon17 | $0.039 \pm 0.015$ | $0.194 \pm 0.039$ | $0.155 \pm 0.049$ |
| | wilds-fmow | $0.530 \pm 0.053$ | $0.588 \pm 0.047$ | $0.059 \pm 0.006$ |
| | wilds-iwildcam | $0.325 \pm 0.022$ | $0.383 \pm 0.017$ | $0.058 \pm 0.009$ |
| | wilds-rxrx1 | $0.901 \pm 0.032$ | $0.927 \pm 0.022$ | $0.025 \pm 0.013$ |
| GMLP | PACS | $0.070 \pm 0.007$ | $0.434 \pm 0.026$ | $0.363 \pm 0.027$ |
| | VLCS | $0.136 \pm 0.011$ | $0.361 \pm 0.016$ | $0.225 \pm 0.022$ |

Continued on next page

| Model | Domain | ID Error | OOD Error | OOD-ID Gap |
|---|---|---|---|---|
| | domain_net | $0.404 \pm 0.012$ | $0.753 \pm 0.013$ | $0.348 \pm 0.017$ |
| | office_home | $0.195 \pm 0.009$ | $0.409 \pm 0.016$ | $0.214 \pm 0.020$ |
| | sviro | $0.034 \pm 0.002$ | $0.260 \pm 0.006$ | $0.226 \pm 0.006$ |
| | terra_incognita | $0.180 \pm 0.016$ | $0.708 \pm 0.010$ | $0.528 \pm 0.021$ |
| | wilds-camelyon17 | $0.041 \pm 0.016$ | $0.214 \pm 0.032$ | $0.173 \pm 0.042$ |
| | wilds-fmow | $0.556 \pm 0.056$ | $0.624 \pm 0.046$ | $0.069 \pm 0.011$ |
| | wilds-iwildcam | $0.338 \pm 0.009$ | $0.329 \pm 0.006$ | $-0.008 \pm 0.011$ |
| | wilds-rxrx1 | $0.972 \pm 0.008$ | $0.977 \pm 0.006$ | $0.005 \pm 0.002$ |
| Mixer | PACS | $0.091 \pm 0.009$ | $0.469 \pm 0.028$ | $0.378 \pm 0.029$ |
| | VLCS | $0.150 \pm 0.011$ | $0.386 \pm 0.017$ | $0.236 \pm 0.022$ |
| | domain_net | $0.433 \pm 0.014$ | $0.785 \pm 0.012$ | $0.352 \pm 0.016$ |
| | office_home | $0.215 \pm 0.011$ | $0.472 \pm 0.016$ | $0.257 \pm 0.020$ |
| | sviro | $0.023 \pm 0.001$ | $0.251 \pm 0.005$ | $0.228 \pm 0.005$ |
| | terra_incognita | $0.172 \pm 0.015$ | $0.737 \pm 0.013$ | $0.565 \pm 0.018$ |
| | wilds-camelyon17 | $0.025 \pm 0.008$ | $0.154 \pm 0.029$ | $0.129 \pm 0.036$ |
| | wilds-fmow | $0.534 \pm 0.042$ | $0.608 \pm 0.038$ | $0.074 \pm 0.005$ |
| | wilds-iwildcam | $0.333 \pm 0.005$ | $0.374 \pm 0.009$ | $0.041 \pm 0.012$ |
| | wilds-rxrx1 | $0.983 \pm 0.006$ | $0.985 \pm 0.005$ | $0.002 \pm 0.001$ |
| ResMLP | PACS | $0.066 \pm 0.006$ | $0.465 \pm 0.026$ | $0.398 \pm 0.026$ |
| | VLCS | $0.145 \pm 0.011$ | $0.395 \pm 0.017$ | $0.250 \pm 0.022$ |
| | domain_net | $0.398 \pm 0.011$ | $0.764 \pm 0.012$ | $0.367 \pm 0.016$ |
| | office_home | $0.203 \pm 0.010$ | $0.453 \pm 0.018$ | $0.250 \pm 0.021$ |
| | sviro | $0.027 \pm 0.001$ | $0.245 \pm 0.005$ | $0.218 \pm 0.005$ |
| | terra_incognita | $0.164 \pm 0.014$ | $0.694 \pm 0.013$ | $0.530 \pm 0.020$ |
| | wilds-camelyon17 | $0.032 \pm 0.012$ | $0.157 \pm 0.015$ | $0.125 \pm 0.025$ |
| | wilds-fmow | $0.544 \pm 0.047$ | $0.613 \pm 0.039$ | $0.069 \pm 0.009$ |
| | wilds-iwildcam | $0.317 \pm 0.006$ | $0.343 \pm 0.009$ | $0.026 \pm 0.008$ |
| | wilds-rxrx1 | $0.949 \pm 0.010$ | $0.959 \pm 0.005$ | $0.010 \pm 0.005$ |
| ResNet50 | PACS | $0.054 \pm 0.005$ | $0.499 \pm 0.032$ | $0.445 \pm 0.033$ |
| | VLCS | $0.137 \pm 0.011$ | $0.325 \pm 0.017$ | $0.189 \pm 0.022$ |
| | domain_net | $0.359 \pm 0.011$ | $0.749 \pm 0.014$ | $0.390 \pm 0.016$ |
| | office_home | $0.180 \pm 0.010$ | $0.431 \pm 0.017$ | $0.251 \pm 0.020$ |
| | sviro | $0.025 \pm 0.001$ | $0.244 \pm 0.005$ | $0.220 \pm 0.006$ |
| | terra_incognita | $0.176 \pm 0.014$ | $0.698 \pm 0.015$ | $0.521 \pm 0.019$ |
| | wilds-camelyon17 | $0.036 \pm 0.013$ | $0.195 \pm 0.045$ | $0.159 \pm 0.052$ |
| | wilds-fmow | $0.512 \pm 0.045$ | $0.579 \pm 0.036$ | $0.067 \pm 0.010$ |
| | wilds-iwildcam | $0.313 \pm 0.015$ | $0.328 \pm 0.007$ | $0.015 \pm 0.010$ |
| | wilds-rxrx1 | $0.894 \pm 0.032$ | $0.925 \pm 0.021$ | $0.030 \pm 0.012$ |
| Swin | PACS | $0.048 \pm 0.006$ | $0.393 \pm 0.027$ | $0.345 \pm 0.027$ |
| | VLCS | $0.121 \pm 0.010$ | $0.350 \pm 0.017$ | $0.230 \pm 0.021$ |
| | domain_net | $0.340 \pm 0.010$ | $0.718 \pm 0.014$ | $0.378 \pm 0.017$ |
| | office_home | $0.152 \pm 0.009$ | $0.351 \pm 0.018$ | $0.199 \pm 0.022$ |
| | sviro | $0.019 \pm 0.001$ | $0.218 \pm 0.005$ | $0.199 \pm 0.005$ |
| | terra_incognita | $0.144 \pm 0.012$ | $0.643 \pm 0.013$ | $0.498 \pm 0.019$ |
| | wilds-camelyon17 | $0.026 \pm 0.010$ | $0.137 \pm 0.020$ | $0.111 \pm 0.028$ |
| | wilds-fmow | $0.481 \pm 0.053$ | $0.548 \pm 0.045$ | $0.067 \pm 0.008$ |
| | wilds-iwildcam | $0.290 \pm 0.004$ | $0.282 \pm 0.004$ | $-0.007 \pm 0.004$ |
| | wilds-rxrx1 | $0.875 \pm 0.042$ | $0.905 \pm 0.027$ | $0.029 \pm 0.016$ |
| ViT-B | PACS | $0.063 \pm 0.008$ | $0.415 \pm 0.027$ | $0.352 \pm 0.027$ |
| | VLCS | $0.137 \pm 0.011$ | $0.341 \pm 0.017$ | $0.205 \pm 0.020$ |
| | domain_net | $0.378 \pm 0.013$ | $0.736 \pm 0.014$ | $0.357 \pm 0.017$ |
| | office_home | $0.163 \pm 0.011$ | $0.367 \pm 0.019$ | $0.204 \pm 0.023$ |
| | sviro | $0.023 \pm 0.001$ | $0.223 \pm 0.006$ | $0.200 \pm 0.006$ |
| | terra_incognita | $0.155 \pm 0.012$ | $0.697 \pm 0.011$ | $0.542 \pm 0.018$ |
| | wilds-camelyon17 | $0.024 \pm 0.007$ | $0.148 \pm 0.022$ | $0.124 \pm 0.029$ |
| | wilds-fmow | $0.495 \pm 0.038$ | $0.565 \pm 0.033$ | $0.070 \pm 0.006$ |
| | wilds-iwildcam | $0.303 \pm 0.006$ | $0.334 \pm 0.021$ | $0.031 \pm 0.017$ |
| | wilds-rxrx1 | $0.951 \pm 0.013$ | $0.966 \pm 0.008$ | $0.015 \pm 0.005$ |

# G   Details on the metrics

We choose a representative set of six metrics used in the robustness literature and describe the details below.

Some quantities are standard metrics, such as *classification error* (top-1 classification error) and *negative log-likelihood (NLL)*. To account for outliers we capped the extreme values of the NLL in the factor analysis as described in B.

We also evaluate the *expected calibration error* (ECE) [104]. The ECE is zero for perfectly calibrated models, i.e., if the predicted probabilities by the model match their true probabilities. We calculate the ECE using 10 bins.

Additionally, two variants of *adversarial classification error* are evaluated by perturbing each test set image using the APGD (Automated Projected Gradient Descent) adversarial attack [124] with an $\ell_2$-attack of size 0.001 and of size 0.02. If not noted otherwise, we only report the mean classification error resulting from both attack sizes.

Lastly we report *Demographic disparity*, which can be interpreted as a measure of invariance. We first split the data into two environments using the method of [77], which maximizes the invariant risk minimization penalty [76]. As the transfer data comes from a single distribution, we would not expect meaningful partitions of the data with systematic differences in the predictions, which we measure with the metric introduced in [103]. Note that, while this metric was introduced to evaluate fairness, it should not be interpreted as such in this paper. The discovered groups may not have any semantic meaning nor fairness implication, so it should not be used to justify that a particular model is fairer than another.

## H   Details on the degrees of freedom and hyperparameter selection

In our study, we evaluate each model listed in Table S4 for all combinations of the hyperparameters listed in Table S3. In order to reduce the overall number of models to train, we first derived good candidates for the learning rate and number of epochs hyperparameters by a larger sweep on a subset of the datasets. For all models we run a large sweep on the the datasets *VLCS-Caltech101*, *OfficeHome-RealWorld* and *DomainNet-Infograph* on the the extended grid of hyperparameters: learning rate ($5e-5, 5e-4, 5e-3, 5e-2$) and the number of training epochs ($3, 10, 100, 1000$). From this we derived the $2 \times 2$ grid of the parameters listed in Table S3. We chose the reduced grid of hyperparameter that lead to the best performance for all models (we made sure that for each model, the best performance is attained by at least one of the hyperparameter combinations in the selected grid). A similar hyperparameter pre-selection strategy was used in [105].

Table S3: In our study, each model is trained for all combinations of hyperparameters listed in this table.

| Training set size | Learning Rate | Train Epochs | Fine-tune | Augmentations |
|:---:|:---:|:---:|:---:|:---:|
| full
few-shot-10
few-shot-100 | $5e-4$
$5e-5$ | 10
100 | Only head
Whole model | No augmentation
RandAugment
AugMix |

Table S4: The list of models used in our study. The pre-trained weights were taken from the PyTorch Image Models package [100] using the displayed model names.

| Model | Timm model name |
|:---|:---|
| Deit | `deit_base_distilled_patch16_224` |
| DenseNet | `densenet169` |
| EfficientNetV2 | `efficientnetv2_rw_s` |
| gMLP | `gmlp_s16_224` |
| MLP-Mixer | `mixer_b16_224` |
| ResMLP | `resmlp_24_224` |
| ResNet50d | `resnet50d` |
| Swin Transformer | `swin_small_patch4_window7_224` |
| Vision Transformer | `vit_base_patch16_224` |

## I   Societal impact, limitations and hardware overview

### I.1   Limitations

Despite best efforts, a large scale experiment like this can never be fully extensive in terms of hyperparameter selection, the choice of model architectures, the evaluated metrics and the overall statistics. We address the limitations by cautious interpretation of the experimental results and the conclusions, see Sections 4 and 5 and particularly the take-away messages therein.

### I.2 Societal impact

This work analyzes how in-distribution metrics relate to out-of-distribution performance. Such questions are often highly relevant when deploying machine learning algorithms to real world systems, since those algorithms get typically trained in specific, possibly idealized environments, which differ from the application environment. Understanding the in- to out-of-distribution generalization properties can therefore easily become relevant for the customer's satisfaction and security. The challenge for improved in- and out-of-distribution generalization is also closely linked to algorithmic fairness [106], which has become a highly relevant societal topic.

### I.3 Computer overview

All $31\mathrm{k}$ experiments were conducted on a cloud hosted cluster using Nvidia T4 GPUs. The aggregated compute time is 17 (GPU-)years.