# OpenReview forum: "Assaying Out-Of-Distribution Generalization in Transfer Learning"
_NeurIPS.cc/2022/Conference — NeurIPS 2022 Accept_

### Official Review · Reviewer_SzPR · 2022-06-28

**Rating:** 7
**Confidence:** 3
**Soundness:** 3 good
**Presentation:** 2 fair
**Contribution:** 3 good

**Summary:**

The setting where training and testing distributions are different in some sense is considered in this work. In particular, a large-scale empirical analysis is carried out with the goal of understanding which evaluation metrics, as measured in-distribution, are predictive of performance out-of-distribution, and what design choices yield improvements under domain shifts. In particular, authors simultaneously considered several datasets to ensure findings are consistent, and not artifacts given by over specialization to particular tasks. In doing so, authors provided evidence indicating that in-distribution prediction performance is a strong predictor of out-of-distribution performance, and additionally reported some indications of design choices yielding good performance in the considered case such as the importance of augmentation approaches.

**Questions:**

Rather than questions, in the following I list suggestion of (non-required) results that would enrich the work in my opinion:

* It would be interesting to understand the effect of doing hyperparameter selection using out-of-distribution data. Do we gain from collecting and labeling some OOD data to do that?

* For section 5, I would suggest authors to include confidence intervals of OOD performance so one could have more (or less) confidence in the recommendations of section 5.

* It would be important to understand how difficult the different tasks are. Perhaps estimating some kind of divergence between ID and OOD samples for each task could serve that purpose. I suspect conclusions might be dependent on that. That is, ID performance will be more indicative of OOD performance for cases with not a large shift.

* To help on the point above, showing pairs of ID/OOD images in the appendix could give an idea of the type of shifts models are facing in each task.

**Limitations:**

* The empirical study focus on the covariate shift setting and it's difficult to assess whether conclusions would hold in more general cases.

* Some of the evaluations, especially in section 5, draw conclusions from comparisons of average performance, which does not provide conclusive evidence.

* It's difficult to get a sense of how hard each task is and whether conclusions are difficulty-dependent.

**Strengths And Weaknesses:**

Strengths:

* A thorough empirical study is carried out covering multiple datasets and model architectures, which renders conclusions broadly applicable under the types of domain shifts the authors considered and within the domain of image classification.

* Results provide evidence that, in the considered setting, in-domain performance is indicative of good out-of-domain performance. This is a highly practical finding which was consistently observed across multiple tasks.

* Overall, conclusions are of high practical relevance and provide directions in terms of designing decisions yielding more robust predictors.

Weaknesses:

* Authors did not clearly discuss the types of domain shifts they accounted for in their evaluations, and seem to consider some unstated assumptions about the distributions faced at testing time. From the discussion and the datasets used for evaluation, it seems it's assumed that only data marginals $p(x)$ shift (i.e., the covariate shift setting), while class conditionals $p(y|x)$ will not change from training to testing time. In fact, that has to be case in order for one to expect to predict OOD performance from ID accuracy, since under conditional shifts counterexamples such as the one discussed in [1] will be such that in- and out-of-distribution performances will be completely uncorrelated.

* While authors did discuss a large amount of related work, to me it seems that some highly relevant related work is not covered. In particular, the first paragraph of the introduction gives the impression that learning under distribution shifts is a problem that only gained attention recently, given failures observed with applications of neural networks. The example of the cow on the beach, for instance, is described as a classical failure case, but it was only discussed a couple of years ago, while the problem of performing risk minimization under domain shifts is under discussion for several years now [2,3]. I would recommend authors to rephrase that paragraph a bit to reflect that.

* The discussion and empirical analysis carried out in section 5 to understand what factors are important in improving OOD performance is limited in that conclusions are drawn based on differences in average performances. Rather, controlled significance tests should've been carried out. That is, for example, the average performances of two distinct architectures are themselves random variables, and the fact that, for a given observation of those, one is greater than the other doesn't imply that this will be often the case. Conclusive tests would, for instance, tell us that 95% of the time one architecture performs better than the other. I suggest a rewording to soften a bit the claims of that section to reflect the fact that results are suggestive rather than conclusive.

* Perhaps a missing factor in the empirical analysis is the difficulty specific of each task. That is to say that not all domain shifts are the same, and as such some of tasks considered by the authors are more difficult than the others (e.g., maybe due to a larger domain shift). It would be insightful to understand how the findings change depending on the "amount" of domain shift.

Summary: All in all, this work presents an insightful empirical evaluation that's useful to the community, but has methodological limitations. More importantly, section 4 doesn't control for task difficulty and in section 5 conclusions are drawn based on comparisons of average performances. Nonetheless, provided that limitations are properly discussed in the text, my opinion is that it should be part of the program. I'll be happy to bump up my score if those limitations are highlighted, the related work is slightly modified to account for relevant missing references, and the setting under consideration (covariate shift?) is formally defined.

[1] Zhao, H., Des Combes, R.T., Zhang, K. and Gordon, G., 2019, May. On learning invariant representations for domain adaptation. In International Conference on Machine Learning (pp. 7523-7532). PMLR.
[2] Ben-David, S., Blitzer, J., Crammer, K. and Pereira, F., 2006. Analysis of representations for domain adaptation. Advances in neural information processing systems, 19.
[3] Ben-David, S., Blitzer, J., Crammer, K., Kulesza, A., Pereira, F. and Vaughan, J.W., 2010. A theory of learning from different domains. Machine learning, 79(1), pp.151-175.

---

> ### Author Response · Authors · 2022-08-02
> **Reply to the reviewer**
>
> > “Authors did not clearly discuss the types of domain shifts they accounted for in their evaluations, and seem to consider some unstated assumptions...”
>
> Most datasets we considered have a strong covariate shift (especially the datasets taken from the DomainBed benchmark).
> However, we can't assume that the covariate shift is the only shift that happens. For instance, we also encounter concept shift and label shift (e.g. in Camelyon17 from the WILDS benchmark, where the labeling process might differer depending on the different hospitals) or subpopulation shift (e.g., FMoW from the WILDS benchmark). In our work we deliberately don’t make any assumption about the type of domain shift since in real-world data we often encounter a combination of different shift types. Some papers focus on specific shift types and analyze them in detail. Our approach, however, is to collect a large variety of dataset pairs to study the different shifts that occur in real datasets.
>
> All our evaluations are agnostic to assumptions on the shift. We only apply standard statistical methods (e.g., computing correlation coefficients between metrics) on the data we have collected. We analyze a large set of ID-OOD dataset pairs from different settings (which we likely include different shift types) and show that we find a clear correlation between some ID and OOD metrics. However, we thank the reviewer to bring up this point and agree that a brief discussion on this topic will improve the paper and we will add it in the final version.
>
> > “While authors did discuss a large amount of related work, to me it seems that some highly relevant related work is not covered. ”
>
> Thank you for your suggestion. We will add the missing related work and will rephrase the paragraph accordingly.
>
> > “Rather, controlled significance tests should've been carried out.”
>
> Thanks for this suggestion. We assessed the statistical significance of the reported gaps in Fig. 4 with a Wilcoxon signed-rank test. We report the p-values in parentheses in Fig. 4. We find that all reported gaps are significant (p-values below 0.05) besides the two gaps in the few-shot-100 setting (ID error and ID ECE).
>
> > “I list suggestion of (non-required) results that would enrich the work in my opinion”
>
> Thank you for those suggestion, we addressed them as outlined below.
>
> > “For section 5, I would suggest authors to include confidence intervals of OOD performance so one could have more (or less) confidence in the recommendations of section 5.”
>
> We have included standard errors to all reported numbers in Table 1, Table S2 and to the new plots in the appendix (Fig. S7, S8). While computing the standard errors, we have noticed that we only used a subset of our data for computing the values in Table 1 in the originally submitted version of the paper. We have updated the numbers accordingly (which didn’t change our conclusions).
>
> > “It would be important to understand how difficult the different tasks are”
>
> Thank you for this input. We have already shown in the paper that some of the previously stated claims don’t hold in general and depend on the task (see section 4.2). We agree that explicitly investigating the dependence of our results on the task difficulty is an interesting experiment. However, computing the difficulty of tasks is an open research question. We agree that computing some sort of distance measure between the ID and OOD distribution is a good starting point, but would not reflect the full picture. Namely, in a direct application, it wouldn’t consider the labels and hence ignore the difficulty of the prediction problem. For now, we choose the following proxy for task difficulty. We compute the (normalized) ID vs. OOD accuracy gap within a task (averaged over the corresponding datasets, all models and fine-tuning methods). We are open to a discussion of what other distance measure could be useful.  The details of our new analysis can be found in section E in the appendix of the revised version. In Fig. S12 in the appendix, we plot the correlation coefficients (between ID metrics and OOD prediction error) as a function of task difficulty. The results are noisy and we don’t observe a clear pattern. Surprisingly, the correlation coefficient seem to be rather independent of the task difficulty.
>
>
> > “showing pairs of ID/OOD images in the appendix could give an idea”
>
> Thank you for the suggestion to make the paper more self-contained, we will add examples of image pairs in the appendix.
>
>
> > “ll be happy to bump up my score if those limitations are highlighted, the related work is slightly modified to account for relevant missing references, and the setting under consideration (covariate shift?) is formally defined.”
>
> We thank the reviewer again for the valuable feedback and hope to have addressed the concerns.

---

> > ### Comment · Reviewer_SzPR · 2022-08-04
> > **Comments on rebuttals**
> >
> > Thank you for your efforts in addressing the raised concerns. I've edited my scores to reflect the modifications/discussion.
> >
> > On the discussion of the settings under consideration, I would highlight that, even if authors are not explicitly relying on a certain set of assumptions, I would strongly recommend for them to discuss in the text the sets of assumptions that are usually considered, and how those would affect their analysis. For instance, strong conditional shift would make it so that it would be impossible to predict OOD performance from ID performance. The fact that there is correlation is evidence that no strong conditional shift is taking place.
> >
> > In terms of a task difficulty measure, the proposed gap seems a good candidate, but just for reference I would point the authors to the H-divergence [1] for the covariate-shift case and to the proposal in [2] for a more general analysis.
> >
> > [1] Ben-David S, Blitzer J, Crammer K, Kulesza A, Pereira F, Vaughan JW. A theory of learning from different domains. Machine learning. 2010 May;79(1):151-75.
> >
> > [2] Zhang G, Zhao H, Yu Y, Poupart P. Quantifying and improving transferability in domain generalization. Advances in Neural Information Processing Systems. 2021 Dec 6;34:10957-70.

---

> > > ### Author Response · Authors · 2022-08-10
> > > **Thanks for your feedback**
> > >
> > > Thank you for your constructive feedback and adjusting your score. We are happy that you've appreciated our efforts. Will include a discussion about the shift types and usual assumption. Also thank you for the references regarding task difficulty, we will take a closer look and try to consider them in our experiments.

---

### Official Review · Reviewer_8h3N · 2022-07-05

**Rating:** 5
**Confidence:** 4
**Soundness:** 3 good
**Presentation:** 3 good
**Contribution:** 3 good

**Summary:**

This paper presents a systematic study of out-of-distribution (OOD) generalization on visual classification. The authors first pointed out the limitations of the current OOD literature, in which the experiments may not be conducted under the same experimental conditions, or the conclusions only hold on a subset of the OOD generalization tasks. To this end, this paper conducts a large-scale empirical evaluations on various kinds of robustness with respect to architecture type, augmentation, fine-tuning strategies and few-shot learning. On top of these evaluations, this paper also derive several interesting findings and recommendations, shedding light on future research.


**Questions:**

- The line spacing of the whole paper is incorrect. The authors should check if any misuse of the NeurIPS LaTex template and reproduce a decent one.


**Limitations:**

The limitations are well discussed in this paper, which mainly concern the selection of hyperparameters.

**Strengths And Weaknesses:**

## Strengths:
- Understanding OOD generalization is an important topic in current machine learning field. This paper provides a systematic view into how ID and OOD metrics correlate, and possible means to improve the OOD robustness.
- This paper is well motivated and clearly written, and the presentation is good. Also, the efforts are well grounded with recent related works.
- The evaluations are comprehensive. The experimental details, as well as the metrics and evaluation factors, are well discussed.

## Weaknesses:
- As an important topic of machine learning, OOD generalization has been studied in many works in recent years, each of which focuses on a few facets of it, as suggested in this paper. While I appreciate the great efforts and useful takeaways, I still think the insights are not significantly greater than former works which only focus on a few facets -- most of the findings of this paper seem not add to our understanding of OOD generalization, besides it is rather complicated and varies with data, models, settings, etc. There could be more constructive suggestions concerning what underlies the generalization, and which architecture design promotes it. The current form of this paper is purely experiment-oriented, while I think some theoretical analysis would be more helpful.
- As suggested in the paper, one conclusion hardly holds in all OOD settings. This may be due to the dataset specialties, and the specific distribution shifts between ID and OOD environments, etc. Similarly, I think it should be more helpful if delving deeper into the exact factors that cause the difference.

---

> ### Author Response · Authors · 2022-08-02
> **Reply to the reviewer**
>
> > “The current form of this paper is purely experiment-oriented, while I think some theoretical analysis would be more helpful... This may be due to the dataset specialties, and the specific distribution shifts between ID and OOD environments, etc. Similarly, I think it should be more helpful if delving deeper into the exact factors that cause the difference.”
>
>
> We indeed conduct an empirical study that focuses on identifying discrepancies and contradicting results in previous papers. We see this as an essential aspect of the scientific research process, which tests the underlying theories and provides new material for future research. For instance, recently, a line of work has emerged around the assumption that there is direct (log-linear) relationship between ID and OOD accuracy (e.g., [1] and [2], which was published after our submission), which we show to be true only in certain specific cases, but not in general. Along with additional contradictions discussed in the paper, we think it is important to show that these assumptions can’t be taken for granted. We expect that our work helps to adjust the assumptions to the real-world setting and also hope that our empirical insights help paving the way to new theoretical insights that are based on realistic assumptions informed by our results.
>
> > “The line spacing of the whole paper is incorrect. The authors should check if any misuse of the NeurIPS LaTex template and reproduce a decent one.”
>
> Thank you, we’ve checked the formatting and corrected it in the revised version.
>
> [1] https://arxiv.org/abs/2107.04649
>
> [2] https://arxiv.org/abs/2206.13089

---

### Official Review · Reviewer_L76V · 2022-07-09

**Rating:** 7
**Confidence:** 4
**Soundness:** 3 good
**Presentation:** 4 excellent
**Contribution:** 3 good

**Summary:**

The authors perform a large-scale investigation of the correlation between ID accuracy, along with various secondary metrics (ID calibration, adversarial robustness, demographic disparity, etc.)  and out-of-distribution (OOD) generalisation.  Models of a wide range of architectures are pre-trained on ImageNet (per convention) and fine-tuned on 172 pairs of ID/OOD datasets pairs which are organised by task. The study builds on a number of other recent works that attempt to characterise the relationship  between ID and OOD performance (e.g. as the latter being (log) linearly predictable by the former) and and how one may validate models while accounting for such problems as underspecification and distribution shift, for which standard validation procedures, where the validation set is derived from the same
distribution as the training set, fail. The results are afforded thorough analysis and several inferences are made that contradict previous findings, while others cement them.





**Questions:**

- Could the fine-tuning strategy/pre-training set to have a significant effect on the results, as suggested by recent works?


**Limitations:**

I am satisfied with the extent that the limitations/ethics of the work have been addressed in both the main text and in Appendix H.

**Strengths And Weaknesses:**

- Overall, I found the paper is well-written and well-structured, and one one provides a meaningful contribution to an area of literature which is still somewhat nebulous -- despite the proliferation in research, works consistent in some respects and inconsistent in others.
- The work is similar in spirit to that of [4] but trades the evaluation of domain generalisation methods for fine-tuning for scale and an evaluation of the utility of secondary metrics for predicting OOD performance (finding little evidence for it when conditioned on ID accuracy).
- Elaborating on the structure, the different components of the study are sectionalised and conclude with a short paragraph of the main takeaways from the experiments in question, making the
paper digestible despite the breadth of the analysis.
- The figures (in tandem with their captions) are generally easy to interpreted and are explained in detail
in the main text.
- The results presented in the main text are further bolstered  by a considerable body of additional results in the supplementary material
- While perhaps not completely in line with the paper's aim, one thing I do feel is neglected, however, is an ablation of the fine-tuning strategy which has been shown to be a critical factor in robust transfer [1, 2, 3].  Particularly, while not explicitly stated, I assume what is meant by  'fine-tuning' is simply end-to-end fine-tuning where the entire model is allowed to be adapted, opposed to linear probing or more exotic techniques (such as bias-tuning [5] or the simply hybrid strategy recently proposed by [2]), which recent works [2, 3] have shown to degrade the robustness inherited from large-scale pre-training.
- Following on from the above, recent work on the interaction between pre-training and OOD robustness, and the role the observed phenomena may have on the current work, warrants some discussion.
- While I understand that the computational prohibitiveness of including multiple replicates, given the already vast number of configurations required to achieve the current results, the lack thereof does challenge the credibility of the conclusions to a degree (particularly apropos of Section 5.1) given the amount  of variation that can be often be chalked up to randomness in the initialisation of the models and in permutations of  the data.
[1] https://arxiv.org/abs/2110.11328

[2] https://openreview.net/forum?id=UYneFzXSJWh

[3] https://arxiv.org/abs/2106.15831

[4] https://arxiv.org/abs/2203.11819

[5] https://arxiv.org/abs/2106.10199

---

> ### Author Response · Authors · 2022-08-02
> **Reply to the reviewer**
>
> > "...ablation of the fine-tuning strategy which has been shown to be a critical factor in robust transfer”
>
> In our experiments we evaluated two fine-tuning strategies: linear probe classifier (i.e., fine-tuning the linear head only) and fine-tuning the full architecture. However, we didn’t include a detailed discussion of the effect of the fine-tuning technique and thank the reviewer for this valuable suggestion. We added a detailed analysis in the appendix C in the revised version of the paper.
>
> We find that fine-tuning the full architecture is usually superior when using the full fine-tuning dataset. Interestingly, when having access to less data (especially in the few-shot-10 setting), we observe that the  linear probe classifier can be better, especially when evaluating on OOD data. This may confirm the idea that changing the last layer only leads to a higher inductive bias by the pre-training data. This is particularly beneficial in low-data regimes when generalizing to OOD data. This insights are in line with previous work (e.g., [1] Fig. 4). We will update our conclusions accordingly.
>
> > “While I understand that the computational prohibitiveness of including multiple replicates, given the already vast number of configurations required to achieve the current results, the lack thereof does challenge the credibility of the conclusions...”
>
> Thank you for bringing this point up; we will discuss this more clearly in the paper. The statistics that we report are typically averaged over multiple datasets (36 ID-ID dataset pair/136 ID-OOD dataset pairs), and/or architectures (9), fine-tuning strategies (2) and augmentation algorithms (3). Hence, every average we report is based on many datapoints. This covers already a large variance and therefore we decided to not further increase the already large number of configurations (in total we ran experiments for 30k+ configurations). To justify this point we computed the standard error for all numbers reported and conducted a significance test for the results in Fig. 4.
>
> [1] https://arxiv.org/abs/2111.07991

---

### Official Review · Reviewer_gFjH · 2022-07-12

**Rating:** 6
**Confidence:** 3
**Soundness:** 3 good
**Presentation:** 3 good
**Contribution:** 2 fair

**Summary:**

This paper experiments on a large number of dataset pairs and models to access the factors for out-of-distribution generalization ability. Importantly, it shows that many OOD performance proxy metrics do not convey more information than the ID accuracy, however, the ID accuracy alone does not explain the OOD performance. Among the proxy metrics are robustness measures and the paper's experiments suggest that adversarial and synthetic corruptions inform OOD performance than calibration, yet all of them are greatly explained by their correlation to ID performance.


**Questions:**

Would the conclusions of the empirical studies in this paper be highly influenced by some specific dataset pairs, for example, those are particularly difficult or have significantly large in-domain and out-of-domain gaps? It would be great if the authors can provide some analyses of the impact of individual datasets.

**Limitations:**

The limitations of this paper have been adequately discussed in Section 5 and the appendix.

**Strengths And Weaknesses:**

This paper conducts experiments on a larger and more comprehensive set of datasets, metrics and models. It shows the effectiveness of ID accuracy for predicting OOD, and shows that many of the previous proxy metrics might only work on a small subset of dataset pairs. While the conclusion from the paper isn't that surprising and the authors did not provide a new metric to better reflect the OOD performance, this paper (and the datasets upon release) would be a good start for a more comprehensively study of this problem.

The paper is generally clear in writing and easy to understand.

---

> ### Author Response · Authors · 2022-08-02
> **Reply to the reviewer**
>
> > “Would the conclusions of the empirical studies in this paper be highly influenced by some specific dataset pairs, for example, those are particularly difficult or have significantly large in-domain and out-of-domain gaps?
>
> We found that dropping individual dataset pairs doesn’t significantly change the reported result, given that we typically average over up to 172 ID-OOD dataset pairs. While compiling our results, we also checked the sensitivity to outliers by systematically replacing the mean with the median in our results and found that it didn't significantly affect our conclusions. However, we do observe clear differences between tasks (which typically consist of multiple datasets, see Table S1 in the appendix). See in particular the task-specific correlation matrices in appendix A.4. To better address your question, we also provide a new plot (Fig. S12) relating our results to the task difficulty, which suggests that there is no clear dependency between the two.

---

### Author Response · Authors · 2022-08-02
**General comments and overview of changes**

We thank all reviewers for the insightful feedback and constructive suggestions to improve the work. Based on the comments we uploaded a revised version of the paper. The main changes are:

1. We included a detailed analysis of the effect of the fine-tuning technique (linear probe classifier vs. fine-tuning the full architecture): section C.1 in the appendix.
2. We added more details on the effect of the augmentation method in section C.2 in the appendix.
3. An analysis of the difficulty of the tasks: section E.1 in the appendix.
4. An analysis of the effect on the difficulty of the task on our main results: section E.2 in the appendix.
5. We conducted a significance test for the performance gaps reported in Fig. 4 and computed standard errors for all other results reported (e.g., Table 1, Table S2, Fig. S7, ...).

Given the limited time for the rebuttal we focused on the changes above. We will continue improving the paper based on the remaining comments/suggestions (as discussed below).

---

### Meta-Review · Area_Chair_62ya · 2022-08-25

**Recommendation:** Accept
**Confidence:** Less certain

**Metareview:**

This paper presents a large-scale empirical study of deep neural networks’ Out-Of-Distribution (OOD) generalization on visual recognition tasks. The main motivation is that previous studies for OOD generalization consider specific types of distributional shifts and are evaluated with specific datasets and architectures, and thus the conclusion drawn from one setting may not generalize to another. To overcome such limitations, the authors perform large-scale experiments under diverse types of OOD data on 172 ID/OOD datasets with multiple architectures. The experimental results show that some of the conclusions drawn from previous small-scale works do not hold in general and may depend on the target tasks, and that the factor that is consistently correlated with the OOD performance is the model’s ID performance.

The reviewers in general agreed that the large-scale empirical study of neural network model’s robustness as well as its results showing the necessity of the large-scale evaluation as useful contributions to the OOD research community, and found the paper well-written and well-structured. This led to an unanimous agreement among the reviewers to accept the paper, but the reviewers (as well as the AC) also had the following concerns:

- The paper is somewhat inconclusive, as the authors simply present the observation that the OOD-robustness of a model may vary from one task to another, without further delving into the problem to see what causes such an issue.

- One consistent observation from the empirical study is that the OOD performance is highly correlated with ID performance, but this is rather trivial; a more meaningful study would be the study of two different models with the same ID performance but with different OOD robustness, and analysis of what contribute to the robustness.

- There is no analysis of the robustness with varying degrees of OOD-ness, although this may be dependent on the type of the task and the dataset, and may be a key to understanding the inconsistency of the robustness across tasks and datasets.

- Overall, the study does not add much to the understanding of OOD robustness due to the above issues, and the study lacks depth.

The authors’ responses mainly focus on clarification and do not really refute the above points, and this looks like a half-baked work, and addressing the above points by undergoing a major revision will definitely strengthen the paper. However, since the large-scale study showing the pitfalls of previous small-scale studies focused on specific types of data distribution still is a valuable contribution that may steer the direction of research and evaluation on the OOD-robustness, a timely publication may be advantageous, and thus I recommend a weak accept for this paper.

**Award:**

No

---

### Decision · Program_Chairs · 2022-09-14

Accept